**Impact of different fertilizers on carbonate weathering in a typical karst area, Southwest China: a field column experiment**

Chao Song [1,2], Changli Liu [1], Guilin Han [2], Congqiang Liu[2]

[1]The Institute of Hydrogeology and Environmental Geology, Chinese Academy of

Geological Sciences, Shijiazhuang, 050803, Hebei, China

[2]School of Water Resources and Environment, China University of Geosciences

(Beijing), Beijing, 100083, China.

Corresponding Author: Chao Song

Email: chao-song@qq.com

Tel/Fax: +86-18931852527

**Abstract:** Carbonate weathering, as a significant vector for the movement of carbon both between and within ecosystems, is strongly influenced by agricultural fertilization, since the addition of fertilizers tends to change the chemical characteristics of soil such as the pH. Different fertilizers may exert a different impact on carbonate weathering, but these discrepancies are as of yet not well-known. In this study, a field column experiment was conducted to explore the response of carbonate weathering to the addition of different fertilizers. We compared 11 different treatments, including a control treatment, using 3 replicates per treatment. Carbonate weathering was assessed by measuring the weight loss of limestone and dolostone tablets buried at the bottom of soil-filled columns. The results show that the addition of urea, $NH_4NO_3$, $NH_4HCO_3$, $NH_4Cl$ and $(NH_4)_2CO_3$ distinctly increased carbonate weathering, which was attributed to the nitrification of $NH_4^+$. The addition of $Ca_3(PO_4)_2$, Ca-Mg-P and $K_2CO_3$ induced carbonate precipitation due to the common ion effect. The addition of $(NH_4)_3PO_4$ and $NaNO_3$ had a relatively little impact on carbonate weathering in comparison to those five $NH_4$-based fertilizers above. The results of $NaNO_3$ treatment raise a new question: the negligible impact of nitrate on carbonate weathering may result in overestimation of the impact of N-fertilizer on $CO_2$ consumption by carbonate weathering at the regional/global scale, if the effects of $NO_3$ and $NH_4$ are not distinguished.

**Keywords:** Carbonate weathering; Column experiment; Nitrogenous fertilizer; Phosphate fertilizer; Southwest China

## 1. Introduction

Carbonate weathering plays a significant role in consumption of atmospheric $CO_2$ (Kump et al., 2000; Liu et al., 2011; Liu et al., 2010). Riverine hydro-chemical composition, such as the ratio of $HCO_3^-$ to $Ca^{2+} + Mg^{2+}$, is usually employed as an indicator to estimate the $CO_2$ consumption by natural carbonate weathering at the regional/global scale (Hagedorn and Cartwright, 2009; Li et al., 2009). However, fluvial alkalinity may also be produced by other processes including the reaction between carbonates and protons derived from: (i) the nitrification of N-fertilizer (Barnes and Raymond, 2009; Gandois et al., 2011; Hamilton et al., 2007; Oh and Raymond, 2006; Perrin et al., 2008; Pierson-wickmann et al., 2009; Semhi and Suchet, 2000; Song et al., 2017; Song et al., 2011; West and McBride, 2005); (ii) sulfuric acid forming in the oxidation of reduced sulfuric minerals (mainly pyrite, $FeS_2$) (Lerman and Wu, 2006; Lerman et al., 2007; Li et al., 2011; Li et al., 2008); (iii) organic acid secreted by microorganisms (Lian et al., 2008); and (iv) acidic soil (such as red soil, yellow soil) (Song et al., 2014; Song et al., 2017). Given that atmospheric $CO_2$ is not a unique weathering agent, differentiating the agent of carbonate weathering is important for the accurate budgeting of net $CO_2$ consumption by carbonate weathering, especially in agricultural areas where mineral fertilizers are used.

The global average annual increase in mineral fertilizer consumption was 3.3 % from 1961 to 1997, and FAO's study predicts a 1 % increase per year until 2030

(FAO, 2000). In China, the consumption of chemical fertilizer increased from 12.7 Mt
in 1980 to 59.1 Mt in 2013 (Fig. 1). The increasing consumption of mineral fertilizers
is a significant disturbance factor in carbonate weathering and the carbon cycle.
Several studies have shown that nitrogen fertilizer additions increased weathering
rates, and also increased the total export of DIC from agricultural watersheds (Barnes
and Raymond, 2009; Gandois et al., 2011; Hamilton et al., 2007; Oh and Raymond,
2006; Perrin et al., 2008; Pierson-wickmann et al., 2009; Probst, 1986; Semhi and
Suchet, 2000; West and McBride, 2005). According to estimates by Probst (1988) and
Semhi et al. (2000), the contribution of N-fertilizers to carbonate dissolution was 30 %
and 12-26 % in two small agricultural carbonate basins in south-western France, the
Girou and the Gers, respectively (tributaries of the Garonne River). In the Garonne
River Basin, which is a large basin (52,000 km$^2$), this contribution was estimated at 6 %
by Semhi et al. (2000). Perrin et al. (2008) estimated that the contribution of
N-fertilizer (usually in form of $NH_4NO_3$) represents up to 5.7-13.4 % and 1.6-3.8 % of
the carbonate dissolution in France and across the world, respectively.
The estimates described above are largely based on calculations that assumed a
single type of fertilizer (e.g. $(NH_4)_2SO_4$, $NH_4NO_3$, or $NH_4Cl$) was used throughout the
whole basin that was considered. However, in actual agricultural practices, different
fertilizers are usually added for different crops. The impact of these fertilizers on
carbonate weathering and riverine chemical composition may be different. In the case
of nitrogenous fertilizer, 100% of $NO_3^-$ produced after the addition $(NH_4)_2SO_4$ and
$NH_4Cl$ is derived from the nitrification of $NH_4^+$, whilst comparatively, it is only 50 %
after the addition of $NH_4NO_3$. Differences in $NO_3^-$ sources may produce a deviation in
the impact of N-fertilizer addition on $CO_2$ consumption by carbonate weathering,
since the addition of different N-fertilizers (e.g. $(NH_4)_2SO_4$, $NH_4NO_3$, $NH_4Cl$, $NaNO_3$
or urea) may result in different contributions to carbonate weathering and relative
products such as $HCO_3^-$, $Ca^{2+}$ and $Mg^{2+}$. For phosphate fertilizer, the coprecipitation
of phosphate ions with calcium carbonate may inhibit carbonate weathering (Kitano et
al., 1978). We assume that the response of carbonate weathering to the addition of
different fertilizers, such as N-fertilizer ($NH_4$ and $NO_3$), P-fertilizer and Ca/Mg
fertilizer, may display differences, which are so far poorly known, but likely
significant. Here we sought to understand the agricultural impact on natural carbonate
weathering, and to accurately evaluate the $CO_2$ consumption via carbonate weathering
in agricultural areas.
The carbonate-rock-tablet test is used to determine the weathering rate of
carbonate rock/mineral from the laboratory to the field (Adams and Post, 1999;
Dreybrodt et al., 1996; Gams, 1981; Gams, 1985; Jiang and Yuan, 1999; Liu and
Dreybrod, 1997; Plan, 2005; Song et al., 2017; Song et al., 2011; Song et al., 2017;
Trudgill, 1975). In the laboratory, the carbonate-rock-tablet is employed to study the
kinetics of calcite dissolution/precipitation (Dreybrodt et al., 1996; Liu and Dreybrod,
1997) and determine the rate of carbonate mineral weathering in the soil column
(Song et al., 2017; Song et al., 2011). In the field, it is also used to observe the rate of
carbonate weathering and estimate $CO_2$ consumption (Jiang, 2013; Jiang and Yuan,
1999; Plan, 2005; Song et al., 2017; Song et al., 2011; Song et al., 2017). Liu (2011)
argued that the carbonate-rock-tablet test may lead to deviations in estimated $CO_2$
consumption by carbonate weathering at the regional/global scale, in cases where
there are insufficient representative data. It is nonetheless a well-established method
for the comparative or simulated experiment (Song et al., 2017; Song et al., 2011;
Song et al., 2017).
A field column experiment that involved embedding carbonate-rock-tablets was
carried out in a typical karst area of southwest China, in order to observe the impact
of different fertilizer additions on carbonate weathering in soil.
**2. Materials and Methods**
**2.1 The study site**
This study was carried out in a typical karst area, namely the Huaxi District of
Guiyang City, Guizhou Province, SW China (26°23′N, 106°40′E, 1094 m ASL).
Guiyang, the capital city of Guizhou Province, is located in the central part of the
province, covering an area from 26°11′00″ to 26°54′20″N and 106°27′20″ to
107°03′00″E (approximately 8,000 km$^2$), with elevations ranging from 875 to 1655 m
ASL. Guiyang has a population of more than 1.5 million people, a wide diversity of
karstic landforms, high elevations and low latitude, with a subtropical warm-moist
climate, and an average annual temperature of 15.3 ℃ and annual precipitation of
1200 mm (Lang, 2006). A monsoonal climate often results in high precipitation during
summer, with much less during winter, although the humidity is often high throughout
most of the year (Han and Jin, 1996). Agriculture is a major land use in order to
produce the vegetables and foods in the suburbs of Guiyang (Liu et al., 2006). The
consumption of chemical fertilizer increased from 150 kg/ha in 1980 to 190 kg/ha in
2013 (GBS, 2014).
**2.2 Soil properties**
The soil used in this column experiment was yellow-brown clay, which was
sampled from the B horizon (below 20 cm in depth) of yellow-brown soil profile from
a cabbage-corn or capsicum-corn rotation plantation in Huaxi District. The soil was
air-dried, ground to pass through a 2-mm sieve, mixed thoroughly and used for the
soil columns. The soil pH ($V_{soil}:V_{water}$ = 1:2.5) was determined by pH meter. The
chemical characteristics of the soil, including organic matter (OM), $NH_4$-N, $NO_3$-N,
available P, available K, available Ca, available Mg, available Fe, and available S
were determined according to the Agro Services International (ASI) method (Hunter,
1984). OM was extracted by using an extracting solution containing 0.2 mol $l^{-1}$
NaOH, 0.01 mol $l^{-1}$ EDTA, 2 % methanol, and 0.005 % Superfloc 127, and
determined by the $K_2CrO_7$ -$H_2SO_4$ oxidation method. $NH_4$-N, $NO_3$-N, available Ca,
and Mg were extracted by 1 mol $l^{-1}$ KCl solution. $NH_4$-N and $NO_3$-N
was determined by colorimetry method, while Ca and Mg were determined by
ICP-AES (inductively coupled plasma atomic emission spectrometer). Available K, P
and Fe were extracted by using an extracting solution containing 0.25 mol $l^{-1}$
$NaHCO_3$, 0.01 mol $l^{-1}$ EDTA, 0.01 mol $l^{-1}$ $NH_4F$, and 0.005 % Superfloc 127. P was
determined by spectrophotometry (colorimetry), and K and Fe were determined by
atomic absorption spectrophotometry. Finally, available S was extracted by 0.1 mol $l^{-1}$
$Ca(H_2PO_4)_2$ and 0.005 % Superfloc 127 and determined by turbidimetric method. The
results are shown in Table 1.
**2.3 Soil column and different fertilization treatments**

In order to test the hypothesis that the impact of different chemical fertilizers on

carbonate weathering may be different, columns (Ø = 20 cm, H= 15 cm) were
constructed from 20 cm diameter polyvinylchloride (PVC) pipe (Fig. 2). A hole (Ø =
2 cm) was placed at the bottom of each column to discharge soil water from the soil
column. A polyethylene net (Ø 0.5 mm) was placed in the bottom of the columns to
prevent soil loss. A 2 cm thick filter layer, including gravel, coarse sand and fine sand,
was spread over the net. Two different carbonate rock tablets were buried in the
bottom of each soil column (Fig. 2). Based on the common kinds of chemical
fertilizers and the main objective of this study, three types (N, P and K fertilizers) of
fertilizer including 10 different fertilizers ($NH_4NO_3$; $NH_4HCO_3$; $NaNO_3$; $NH_4Cl$;
$(NH_4)_2CO_3$; $Ca_3(PO_4)_2$; $(NH_4)_3PO_4$; fused calcium-magnesium phosphate; Urea and
$K_2CO_3$ fertilizer) were involved in this study. As a result, eleven fertilization
treatments including control treatment, each with three replicates, were set up in the
field column experiment. The local practical rate of fertilizer application is
approximate 160 kg N $\cdot$ ha$^{-1}$ of N fertilizer, 150 kg $P_2O_5$ ha$^{-1}$ of P fertilizer and 50 kg
K $\cdot$ ha$^{-1}$ of K fertilizer. In order to short the time of this experiment and make the
experimental results distinct, the added amount of 10 fertilizers was increased to: (1)
control without fertilizer (CK); (2) 43g $NH_4NO_3$ fertilizer (CF); (3) 85g $NH_4HCO_3$
fertilizer (NHC); (4) 91g $NaNO_3$ fertilizer (NN); (5) 57g $NH_4Cl$ fertilizer (NCL); (6)
51g $(NH_4)_2CO_3$ fertilizer (NC); (7) 52g $Ca_3(PO_4)_2$ fertilizer (CP); (8) 15g $(NH_4)_3PO_4$
fertilizer (NP); (9) 44g fused calcium-magnesium phosphate fertilizer (Ca-Mg-P); (10)
32g Urea fertilizer (U); and (11) 10g $K_2CO_3$ fertilizer (PP). An aliquot of 6 kg of soil
was weighed (bulk density = 1.3 g/cm$^3$), mixed throughly with one of the above
fertilizers, and filled into its own column. This process was repeated for all three
replicates of the 11 fertilizer treatments. The soil columns were labelled and placed
orderly (see Fig. 2b) at the field experiment site in Huaxi District, Guiyang for a
whole year.

**2.4 The rate of carbonate weathering**

Two different kinds of carbonate rock tablets (2 cm $\times$ 1 cm $\times$ 0.5 cm in size) were
placed in the bottom of each soil column to examine the rate of carbonate weathering
in the soil. The two different kinds of carbonate rock collected from the karst area of
Huaxi District were: (1) limestone with 60-65 % micrite, 30-35 % microcrystalline
calcite, and 2-3 % pyrite; and (2) dolostone with 98-99 % fine crystalline dolomite, 1 %
pyrite, and trace quantities organic matter. All the tablets were heated at 80 ℃ for 4
hours, weighed on a 1/10000 electronic balance in the laboratory, labeled by tying a
label with fishing line, and then buried at the bottom of each soil column. After a
whole year, the tablets were removed carefully, rinsed, baked and weighed.
The amount of weathering ($A_w$), the ratio of weathering ($R_w$) and the rate of
weathering ($R_{aw}$) for limestone and dolomite were calculated according to the weight
difference of the tablets using the following formulas:

$$A_w = (W_i - W_f) \tag{1}$$

$$R_w = (W_i - W_f)/W_i \tag{2}$$

$$R_{aw} = (W_i - W_f)/(S*T) \tag{3}$$

where $W_i$ is the initial weight of the carbonate-rock-tablet, $W_f$ is the final weight, S is
the surface area of carbonate rock tablet, and T is the length of the experimental
period.

**2.5 Statistical analysis**

Statistical analysis was performed using IBM SPSS 20.0 (Statistical Graphics
Corp, Princeton, USA). All results of carbonate weathering were reported as the
means ± standard deviations (SD) for the three replicates.
**3. Results**
3.1 Weathering rate of carbonate under different fertilized treatments
The R$w$ and R$aw$ of limestone and dolostone are listed in Table 2. The results
show that the R$w$ of limestone under urea, $NH_4NO_3$, $NH_4Cl$, $(NH_4)_2CO_3$ and
$NH_4HCO_3$ treatments were 8.48 ± 0.96, 6.42 ± 0.28, 5.54 ± 0.64, 4.44 ± 0.81 and 4.48
± 0.95 ‰, respectively, significantly greater than that under the control treatment 0.48
± 0.14 ‰ (see Fig. 3). In addition, the observed R$w$ of dolostone were 6.59 ± 0.67,
5.30 ± 0.87, 4.77 ± 0.78, 4.94 ± 1.91 and 3.22 ± 0.87 ‰ respectively, under these
same five fertilization treatments, in contrast to -0.31 ± 0.09 ‰ in the control
treatment. This clearly demonstrates that the addition of these five fertilizers increased
the rate of carbonate weathering.
The remaining treatments made differences in the R$w$ and R$aw$ of limestone and
dolostone in comparison to the control treatment (Table 2), but the differences were
much smaller than the treatments with those five fertilizers as mentioned above (Fig.
3). In the $(NH_4)_3PO_4$ treatment, the R$w$ were only 1.08 ± 0.34 ‰, and 0.75 ± 0.21‰
for limestone and dolomite, respectively, while the R$aw$ were 4.00 ± 1.15 g m$^{-2}$ a$^{-1}$
and 1.00 ± 1.01 g m$^{-2}$ a$^{-1}$ for limestone and dolomite, respectively. The R$w$ and R$aw$
in the $NaNO_3$ treatments showed differences with the control treatment. The values,
however, are much less than those under the five $NH_4$-based fertilizers mentioned
above, exhibiting little effect of the $NaNO_3$ fertilizer addition on carbonate weathering
(see Table 2 and Fig. 3). Except for the R$w$ of limestone approaching zero in the
$Ca_3(PO_4)_2$ treatment, all the values of R$w$ and R$aw$ in Ca-Mg-P, $K_2CO_3$ and $Ca_3(PO_4)_2$
treatments showed negative values. This indicates that the addition of Ca-Mg-P,
$K_2CO_3$ and $Ca_3(PO_4)_2$ fertilizers led to precipitation at the surface of the carbonate
mineral.
3.2 Comparison of limestone of dolomite
Fig. 3 shows that, on the whole, the ratios of dolostone weathering are smaller
than those of limestone weathering except for the $(NH_4)_2CO_3$ treatment,
demonstrating that dolostone weathers more slowly than limestone under fertilization
effects.
In Fig. 4, we plotted the R$w$ of limestone vs. dolostone tablets in a linear
correlation diagram, in order to compare the weathering responses of limestone with
dolostone. The results show that the R$w$ of limestone and dolostone exhibit a high
positive correlation ($R^2$=0.9773; see Fig. 4), indicating that the weathering of
limestone and dolostone are similar under different treatments. Thus, we will explain
the results in terms of carbonates, rather than separately discussing the individual
dolostone and limestone.
**4. Discussion**
**4.1 Kinetics of carbonate dissolution/precipitation: controlling factors**
Experimental studies of carbonate dissolution kinetics have shown metal
carbonate weathering usually depends upon three parallel reactions occurring at the
carbonate interface (Chou et al., 1989; Plummer et al., 1978; Pokrovsky et al., 2009):
$$MeCO_3 \leftrightarrow Me^{2+} + CO_3^{2-} \tag{4}$$
$$MeCO_3 + H_2CO_3 \leftrightarrow Me^{2+} + 2HCO_3^- \tag{5}$$
$$MeCO_3 + H^+ \leftrightarrow Me^{2+} + HCO_3^- \tag{6}$$
where Me = Ca, Mg. As Eq. (5) describes, atmospheric/soil $CO_2$ is usually considered
to be the natural weathering agent of carbonate. In watersheds with calcite- and
dolomite-containing bedrock, $H_2CO_3$ formed in the soil zone usually reacts with
carbonate minerals, resulting in dissolved Ca, Mg, and $HCO_3^-$ as described in Eq. (5)
(Andrews and Schlesinger, 2001; Shin et al., 2014). Although it has been proven that
the reaction of carbonate dissolution is mainly controlled by the amount of rainfall
(Amiotte Suchet et al., 2003; Egli and Fitze, 2001; Kiefer, 1994), in this study, we
consider that the effect of rainfall is equal in each soil column, and hence is
disregarded as a controlling factor in weathering rate differences among these
treatments. In theory, the fertilizers could stimulate bacteria, which may increase
respiration and $CO_2$ concentrations in the soil, as a result, probably enhance
carbonate weathering as Eq. (5). However, Eq. (6) suggests that the proton from other
origins, such as the nitrification processes of $NH_4^+$, as mentioned in the Introduction
section, can play the role of weathering agent in agricultural areas. In this study, the
urea, $NH_4NO_3$, $NH_4HCO_3$, $NH_4Cl$, and $(NH_4)_2CO_3$ amendments increased (10 to
17-fold) the natural weathering rate from 2.00 g $m^{-2}$ $a^{-1}$ for limestone tablets in the
control treatment (Table 2). Thus, these increases are strongly related to the effect of
proton release from the nitrification of $NH_4^+$. In contrast, carbonate precipitation will

occur due to the backward reaction of Eq. (5) in the following cases: (1) the degassing of dissolved $CO_2$ due to dramatic changes in the parameters of the $CO_2$ system (such as T, pH, $pCO_2$, etc); (2) soil evapotranspiration; or (3) the common ion effect: the increase of $Ca^{2+}$, $Mg^{2+}$ or $CO_3^{2-}$ in a weathering-system with equilibrium between water and calcite (Calmels et al., 2014; Dreybrodt, 1998).

**4.2 Main reactions and effects in different treatments**

The main reactions and effects of every treatment in this study are listed in Table 3.

**(1) Nitrification in $NH_4$-fertilizer: $NH_4NO_3$, $NH_4HCO_3$, $NH_4Cl$, $(NH_4)_2CO_3$ and urea**

In urea $(CO(NH_2)_2)$ treatment, the enzyme urease rapidly hydrolyzes the urea-N to $NH_4^+$ ions (Eq. (7)) when urea is applied to the soil (Soares et al., 2012).

$$CO(NH_2)_2 + 3H_2O \rightarrow 2NH_4^+ + 2OH^- + CO_2 \tag{7}$$

Although the study of Singh et al. (2013) showed that part of $NH_4^+$ may be lost as ammonia ($NH_3$) and subsequently as nitrous oxide ($N_2O$) (Singh et al., 2013), the remaining ammonium ($NH_4^+$) is mainly oxidized during nitrification in soil by autotrophic bacteria, such as Nitrosomonas, resulting in nitrite $NO_2^-$ and $H^+$ ions. Nitrite is in turn oxidized by another bacterium, such as Nitrobacter, resulting in nitrate ($NO_3^-$) (Eq. (8)) (Perrin et al., 2008).

$$NH_4^+ + 2O_2 \rightarrow NO_3^- + H_2O + 2H^+ \tag{8}$$

The protons ($H^+$) produced by nitrification can be neutralized in two ways:

(i)   either by exchange process with base cations in the soil exchange complex

(Eq. (9))        $Soil - Ca + 2H^+ \rightarrow Soil - 2H^+ + Ca^{2+}$        (9)

or (ii) via carbonate mineral dissolution (Eq.(10))
$$Ca_{(1-x)}Mg_xCO_3 + H^+ \rightarrow (1-x) Ca^{2+} + xMg^{2+} + HCO_3^- \tag{10}$$
Consequently, by combining Eq. (8) and Eq. (10), carbonate weathering by
protons produced by nitrification can be expressed as (Eq. 11) (See details in Perrin et
al., 2008 and Gandois et al., 2011).
$$2Ca_{(1-x)}Mg_xCO_3 + NH_4^+ + 2O_2 \rightarrow 2(1-x) Ca^{2+} + 2xMg^{2+} + NO_3^- + H_2O + 2HCO_3^-$$
$$\tag{11}$$
As discussed above, provided that the loss as ammonia ($NH_3$) and nitrous oxide
($N_2O$) after hydrolyzation is disregarded in this study, the final equation of carbonate
weathering in $NH_4NO_3$, $NH_4HCO_3$, $NH_4Cl$, $(NH_4)_2CO_3$ and urea treatments will be as
follows, respectively:
$$2Ca_{(1-x)}Mg_xCO_3 + NH_4NO_3 + 2O_2 \rightarrow 2(1-x) Ca^{2+} + 2xMg^{2+} + 2NO_3^- + H_2O +$$
$$2HCO_3^- \tag{12}$$
$$2Ca_{(1-x)}Mg_xCO_3 + NH_4HCO_3 + 2O_2 \rightarrow 2(1-x) Ca^{2+} + 2xMg^{2+} + NO_3^- + H_2O +$$
$$3HCO_3^- \tag{13}$$
$$2Ca_{(1-x)}Mg_xCO_3 + NH_4Cl + 2O_2 \rightarrow 2(1-x) Ca^{2+} + 2xMg^{2+} + NO_3^- + Cl^- + H_2O +$$
$$2HCO_3^- \tag{14}$$
$$3Ca_{(1-x)}Mg_xCO_3 + (NH_4)_2CO_3 + 4O_2 \rightarrow 3(1-x) Ca^{2+} + 3xMg^{2+} + 2NO_3^- + 2H_2O +$$
$$4HCO_3^- \tag{15}$$
$$3Ca_{(1-x)}Mg_xCO_3 + CO(NH_2)_2 + 4O_2 \rightarrow 3(1-x) Ca^{2+} + 3xMg^{2+} + 2NO_3^- + 4HCO_3^-$$
$$\tag{16}$$
**(2) No effect of $NO_3$-fertilizer treatment: $NaNO_3$ treatment**
In the $NaNO_3$ treatment, the reaction occurs according to Eq. (17), indicating that
the addition of $NO_3$-fertilizer does not significantly influence carbonate weathering.
$$Ca_{(1-x)}Mg_xCO_3 + NaNO_3 + CO_2 + H_2O \rightarrow (1-x) Ca^{2+} + xMg^{2+} + Na^+ + NO_3^- +$$
$2HCO_3^-$ (17)

**(3) Common ion effect: $K_2CO_3$ treatment**

In the $K_2CO_3$ treatment, $CO_3^{2-}$ and $HCO_3^-$ will be produced after the addition of
$K_2CO_3$ according to Eq. (18), hence resulting in carbonate precipitation as described
in Eq. (19), due to the common ion effect.
$$K_2CO_3 + H_2O \rightarrow 2K^+ + HCO_3^- + OH^-$$ (18)
$$(1-x) Ca^{2+} + xMg^{2+} + 2HCO_3^- \rightarrow Ca_{(1-x)}Mg_xCO_3 + CO_2 + H_2O$$ (19)

**(4) Complex effects: Nitrification versus inhibition effect of $PO_4$ in $(NH_4)_3PO_4$ treatments**

In the $(NH_4)_3PO_4$ treatment, the reaction of carbonate weathering will occur
according to Eq. (11) due to the nitrification of $NH_4^+$ ionized from the $(NH_4)_3PO_4$
fertilizer. The $PO_4^{3-}$ anion will exert an inhibition to calcite dissolution (Kitano et al.,
1978), as calcium orthophosphate (Ca-P) precipitation is produced on the surface of
calcite after the addition of $PO_4^{3-}$ in soil (reaction: $Ca + PO_4 \rightarrow Ca-P$), resulting in
inhibition of the calcite dissolution (Alkattan et al., 2002; Berner and Morse, 1974;
Raistrick, 1949).

**(5) Complex effects: Common ion effect versus inhibition effect of $PO_4$ in $Ca_3(PO_4)_2$ and Ca-Mg-P treatments**

In the $Ca_3(PO_4)_2$ and Ca-Mg-P treatments, on the one hand, $Ca_{(1-x)}Mg_xCO_3$ is
produced when the concentrations of $Ca^{2+}$ (or/and $Mg^{2+}$) increases according to Eq.
(19). On the other hand, the inhibition effect of phosphate will cause calcium
phosphate precipitation to be produced on the surface of carbonate minerals after the
addition of P in soil (reaction: $Ca + PO_4 \rightarrow Ca-P$), resulting in inhibition the carbonate
precipitation (Alkattan et al., 2002; Burton and Walter, 1990; Giannimaras and
Koutsoukos, 1987; House, 1987; Ishikawa and Ichikuni, 1981; Lin and Singer, 2006;
Mucci, 1986; Reddy, 1977).

**4.3 Difference between $NH_4^+$ and $NO_3^-$ in impacts on carbonate weathering and**

**implication on the estimation of $CO_2$ consumption**
In order to further compare the differences between $NH_4^+$ and $NO_3^-$ effects on
carbonate weathering, the initial molar amounts of fertilizer-derived $NH_4$ per unit in
every treatment were calculated, and are listed in Table 4. The results show that the
amount of $NH_4^+$ hydrolyzed from added urea is 1.06 mole, while $NH_4^+$ ionized from
added $NH_4NO_3$, $NH_4HCO_3$, $NH_4Cl$, $(NH_4)_2CO_3$ and $(NH_4)_3PO_4$ is 0.54 mole, 1.08
mole, 1.07 mole, 1.06 mole, and 0.03 mole, respectively (Table 4). The R$w$ of
limestone tablets and the initial amount of $NH_4^+$ per treatment are plotted in Fig. 5. A
distinct relationship between them is observed, in that the R$w$ values in $NH_4NO_3$,
$NH_4HCO_3$, $NH_4Cl$, $(NH_4)_2CO_3$ and urea treatments are larger than in the control
treatment, where the initial amount of $NH_4^+$ yields similar results (Fig. 5). This
suggests that carbonate weathering in $NH_4NO_3$, $NH_4HCO_3$, $NH_4Cl$, $(NH_4)_2CO_3$ and
urea treatments are mainly attributed to the dissolution reaction described as Eq. (11).
This process of carbonate weathering by protons released from nitrification has been
demonstrated by many studies, from the laboratory to the field (Barnes and Raymond,
2009; Bertrand et al., 2007; Biasi et al., 2008; Errin et al., 2006; Gandois et al., 2011;
Hamilton et al., 2007; Oh and Raymond, 2006; Perrin et al., 2008; Semhi and Suchet,
2000; Song et al., 2017; Song et al., 2011; West and McBride, 2005). We have noted
that the R$w$ values in $NH_4HCO_3$ and $(NH_4)_2CO_3$ treatments are less than half those in
urea treatment despite adding the same amount of fertilizer-derived $NH_4$
(approximately 1.07 mole). This is probably because the two fertilizers, $NH_4HCO_3$
and $(NH_4)_2CO_3$ as a typical weak acid and weak base salt, are easier to decompose
and produce $NH_3$ and $CO_2$ gases according to Eq. (20) and (21) (Trypuc and
Kielkowska, 1996), resulting in amounts of fertilizer-derived $NH_4$ that are lower than
1.07 moles.
$$NH_4HCO_3 \rightarrow NH_3 \uparrow + H_2O + CO_2 \uparrow \qquad (20)$$
$$(NH_4)_2CO_3 \rightarrow 2NH_3 \uparrow + H_2O + CO_2 \uparrow \qquad (21)$$
The $Aw$ and $Rw$ in the $(NH_4)_3PO_4$ treatment, unlike in other $NH_4$-fertlizer
treatments, do not show a significant increase compared to the control treatment,
which is not only owing to the low amount of added $NH_4^+$ in $(NH_4)_3PO_4$ treatment
(0.3 mole; see Table 4), but also the inhibition of phosphate (Chien et al., 2011; Wang
et al., 2012). After the addition of $(NH_4)_3PO_4$ in soil, calcium orthophosphate (Ca-P)
precipitation will form on calcite surfaces, which is initiated with the aggregation of
clusters leading to the nucleation and subsequent growth of Ca-P phases, at various
pH values and ionic strengths relevant to soil solution conditions (Chien et al., 2011;
Wang et al., 2012).
There is no significant different between the $Rw$ in the $NaNO_3$ treatment
compared to the control treatment, indicating that the addition of $NO_3$-fertilizer does
not significantly influence carbonate weathering.
A notable issue herein is that the $NaNO_3$ treatment produces the same amount of
$NO_3^-$ (1.07 mole) as other $NH_4$ fertilizer ($NH_4NO_3$, $NH_4HCO_3$, $NH_4Cl$, $(NH_4)_2CO_3$
and urea), but it fails to impact carbonate weathering, which raises a new problem. Eq.
(5), usually considered as an expression for the natural weathering process of
carbonate, is an important reaction in understanding the kinetics of carbonate
dissolution in carbonate-dominated areas, where the molar ratio of $HCO_3^-$ and $Me^{2+}$ in

the river is usually used as an indicator to make estimates of $CO_2$ consumption by

carbonate weathering at the regional/global scale (Hagedorn and Cartwright, 2009; Li

et al., 2009). In agricultural areas, the relationship between $(Ca+Mg)/HCO_3^-$ and $NO_3^-$

is usually employed to estimate the contribution of N-fertilizer to riverine $Ca^{2+}$, $Mg^{2+}$,

and alkalinity (Etchanchu and Probst, 1988; Jiang, 2013; Jiang et al., 2009; Perrin et

al., 2008; Semhi and Suchet, 2000). In these studies, the nitrification described in Eq.

(8) is usually considered as the unique origin of $NO_3^-$. According to the results of the

$NaNO_3$ treatment in this study, the contribution of protons from nitrification to

carbonate weathering may be overestimated, if anthropogenic $NO_3^-$ is neglected, since

the anthropogenic $NO_3^-$ does not release the proton described in Eq. (8). For $NH_4NO_3$

fertilizer, Eq. (12) shows that two moles of $Ca^{2+}+Mg^{2+}$, $NO_3^-$, and $HCO_3^-$ will be

produced when one mole $NH_4NO_3$ reacts with 2 moles of carbonate, where only half

of the $NO_3^-$ originates from nitrification described as Eq. (8). This will result in an

overestimation of the contribution of nitrification to carbonate weathering, and thus

thereby mislead the estimation of $CO_2$ consumption.

At regional scales, if different fertilizers are simultaneously added to an

agricultural area, the estimation of $CO_2$ consumption by carbonate weathering might

become more complicated, since the mole ratios of $Ca+Mg$, $HCO_3^-$, and/or $NO_3^-$

between different fertilization treatments are different (see Table 3). Thus, the related

anthropogenic inputs (e.g. $Ca+Mg$, $NH_4$, $NO_3^-$, $HCO_3^-$, etc.) need to be investigated to

more accurately estimate the impact of fertilization on carbonate weathering and its

$CO_2$ consumption.

**4.4 The comparison with other studied results**

The R$w$ and R$aw$ of limestone in the control treatment in this study were 0.48 ‰

and 2.00 g m$^{-2}$ a$^{-1}$, respectively. These are generally consistent with observations of

0.51-32.97 g $m^{-2}$ $a^{-1}$ (for R$aw$) in Nongla, Guangxi, a karst area of Southwestern
China (Zhang, 2011), and with the results of 0.05-5.06 ‰ (for R$w$) and 1.08-136.90
g $m^{-2}$ $a^{-1}$ (for R$aw$) from the north slope of the Hochschwab Massif in Austria (Plan,
2005), as determined using the limestone tablet method. But the R$aw$ of 2.00 g $m^{-2}$ $a^{-1}$
is lower than the results of 7.0-63.5 g $m^{-2}$ $a^{-1}$ for R$aw$ from Jinfo Mountain in
Chongqing, China (Zhang, 2011). These differences in carbonate weathering are
mainly attributed to the different types of carbonate rock tablets, climate,
micro-environments of soil, etc. The R$aw$ of limestone in the N-fertilizer treatments
are 20.57-34.71 g $m^{-2}$ $a^{-1}$, similar to the weathering rate of carbonate in an orchard
(32.97 g $m^{-2}$ $a^{-1}$) at Nongla, Manshan, Guangxi, China, which usually involves
fertilization activities.

At larger scales, such as watersheds, the weathering rate is usually estimated by

using the riverine hydro-chemical method, which is inconsistent with the results from
the carbonate-rock-tablet test. Zeng et al. (2014) estimate that the carbon sink
intensity calculated by the carbonate-rock-tablet test is only one sixth of that
estimated using the riverine hydro-chemical method (Zeng et al., 2014). The results
from Semhi et al. (2000) show that weathering rates of carbonate rock using riverine
hydro-chemical method are approximately 77.5 g $m^{-2}$ $a^{-1}$ and 50.4 g $m^{-2}$ $a^{-1}$ in the
upstream and downstream, respectively, of the Garonne river, France, which are
approximately 25-35 times greater than that in the control treatment (2.00 g $m^{-2}$ $a^{-1}$
for natural weathering rate) and 2-3 times greater than in the N-fertilizer treatment
(20.57-34.71 g $m^{-2}$ $a^{-1}$ for anthropic weathering rate) in this study. The global natural
weathering rate of carbonate reported by Amiotte Suchet et al. (2003) is 47.8 g $m^{-2}$ $a^{-1}$,
which is much higher than that we observed. Thus, we conclude that it is difficult to
compare the results from the carbonate-rock-tablet test and the riverine

hydro-chemical method. The carbonate-rock-tablet test is suitable for research on the comparative or simulated experiments, while the riverine hydro-chemical method is appropriate for regional investigations and estimations. According to the estimation from Yue et al. (2015), the enhanced $HCO_3^-$ flux due to nitrification of $NH_4^+$ at Houzhai catchment of Guizhou Province would be $3.72 \times 10^5$ kg C/year and account for 18.7 % of this flux in the entire catchment (Yue et al., 2015). This is similar to estimates from other small agricultural carbonate basins (12–26 %) in southwest France.

**5. Conclusions**

The impact of the addition of different fertilizers ($NH_4NO_3$, $NH_4HCO_3$, $NaNO_3$, $NH_4Cl$, $(NH_4)_2CO_3$, $Ca_3(PO_4)_2$, $(NH_4)_3PO_4$, Ca-Mg-P, urea, and $K_2CO_3$) on carbonate weathering was studied in a field column experiment using carbonate-rock-tablets. The amount of weathering and the ratio of weathering of carbonate rock tablets showed that the addition of urea, $NH_4NO_3$, $NH_4HCO_3$, $NH_4Cl$, and $(NH_4)_2CO_3$ distinctly increased carbonate weathering, which was attributed to the nitrification of $NH_4^+$, while the addition of $Ca_3(PO_4)_2$, Ca-Mg-P and $K_2CO_3$ induced carbonate precipitation due to the common ion effect. The addition of $(NH_4)_3PO_4$ and $NaNO_3$ had a relatively little impact on carbonate weathering, where the former can be attributed to the low added amount of $(NH4)_3PO_4$, and may be related to the inhibition of phosphate, while the latter seemed to raise a new question. The question is:the minor impact of nitrate on carbonate weathering may result in the overestimation of the impact of N-fertilizer on $CO_2$ consumption by carbonate weathering at the regional/global scale, if the effects of $NO_3$ and $NH_4$ are not distinguished. Thus, the

related anthropogenic inputs (e.g. Ca+ Mg, $NH_4$, $NO_3^-$, $HCO_3^-$, etc.) need to be investigated to more accurately estimate the impact of fertilization on carbonate weathering and its consumption of $CO_2$ (Perrin et al., 2008; Semhi and Suchet, 2000).

**6. Acknowledgements**

This study was supported jointly by the Basic Science Research Fund from the Institute of Hydrogeology and Environmental Geology (Grant No. SK201208), and the Chinese National Natural Science Foundation (No. 41403107 and No. 41325010).

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

Table 1 Chemical composition of soil

| Parameter | Unit | Values |
|---|---|---|
| pH | - | 6.94 |
| Content of particles <0.01mm | % | 74 |
| Content of particles <0.001mm | % | 45 |
| Organic matter | % | 0.99 |
| $NH_4^+$-N | mg/kg | 339.87 |
| $NO_3^-$-N | mg/kg | 569.05 |
| Available P | mg/kg | 8.18 |
| Available K | mg/kg | 56.88 |
| Available Ca | mg/kg | 3041.06 |
| Available Mg | mg/kg | 564.83 |
| Available S | mg/kg | 100.72 |
| Available Fe | mg/kg | 24.41 |



Table 2 Carbonate weathering under different fertilizer treatments

| Treatment | Limestone | | Dolostone | |
|---|---|---|---|---|
| | $Rw$ / ‰ | $Raw$ / g m$^{-2}$ a$^{-1}$ | $Rw$ / ‰ | $Raw$ / g m$^{-2}$ a$^{-1}$ |
| Control | 0.48 ±0.14 | 2.00 ±0.58 | -0.31±0.09 | -1.57 ±0.86 |
| $NH_4NO_3$ | 6.42 ±0.28 | 24.86 ±2.01 | 5.30±0.87 | 20.57 ±1.15 |
| $NH_4HCO_3$ | 4.44 ±0.81 | 21.00 ±3.45 | 3.22±0.87 | 13.71 ±3.88 |
| $NaNO_3$ | 0.86 ±0.17 | 4.43 ±1.73 | 0.53 ±0.26 | 3.14 ±1.73 |
| $NH_4Cl$ | 5.54 ±0.64 | 21.29 ±2.45 | 4.77 ±0.78 | 18.71 ±0.86 |
| $(NH_4)_2CO_3$ | 4.48 ±0.95 | 20.57 ±4.46 | 4.94 ±1.91 | 26.57 ±7.62 |
| $Ca_3(PO_4)_2$ | 0.01 ±0.04 | 0.43 ±0.86 | -0.55 ±0.25 | -1.86 ±1.29 |
| $(NH_4)_3PO_4$ | 1.08 ±0.34 | 4.00 ±1.15 | 0.75 ±0.21 | 1.00 ±1.01 |
| Ca-Mg-P | -0.31 ±0.12 | -1.86 ±0.43 | -0.97 ±0.38 | -3.14 ±0.72 |
| Urea | 8.48 ±0.96 | 34.71 ±4.32 | 6.59 ±0.67 | 26.43 ±2.73 |
| $K_2CO_3$ | -0.26 ±0.15 | -1.14 ±0.58 | -0.59 ±0.15 | -2.57 ±0.43 |

$Rw$ - the ratio of carbonate weathering; $Raw$ - the rate of carbonate weathering; $Rw$ =1000 ($Wi$-$Wf$)/Wi
and $Raw$ = ($Wi$-$Wf$)/(S*T), where $Wi$ is the initial weight of the carbonate rock tablet, and $Wf$ is the
final weight. S is the surface area of carbonate rock tablet (here, we used S = 7 cm$^2$ for every tablet),
and T is the experiment period. Values are reported as means ±standard deviations, n = 3.

Table 3 The main reaction and effects in the 11 fertilizer treatments

| Treatment | Main reactions and effects |
|---|---|
| 1. Control | $Ca_{(1-x)}Mg_xCO_3 + CO_2 + H_2O \rightarrow (1-x)\ Ca^{2+} + xMg^{2+} + 2HCO_3^-$ |
| 2. $NH_4NO_3$ | $2Ca_{(1-x)}Mg_xCO_3 + NH_4NO_3 + 2O_2 \rightarrow 2(1-x)\ Ca^{2+} + 2xMg^{2+} + 2NO_3^- + H_2O + 2HCO_3^-$ |
| 3. $NH_4HCO_3$ | $NH_4HCO_3 \rightarrow NH_3 \uparrow + H_2O + CO_2 \uparrow$<br>$2Ca_{(1-x)}Mg_xCO_3 + NH_4HCO_3 + 2O_2 \rightarrow 2(1-x)\ Ca^{2+} + 2xMg^{2+} + NO_3^- + H_2O + 3HCO_3^-$ |
| 4. $NaNO_3$ | $Ca_{(1-x)}Mg_xCO_3 + NaNO_3 + CO_2 + H_2O \rightarrow (1-x)\ Ca^{2+} + xMg^{2+} + Na^+ + NO_3^- + 2HCO_3^-$ |
| 5. $NH_4Cl$ | $2Ca_{(1-x)}Mg_xCO_3 + NH_4Cl + 2O_2 \rightarrow 2(1-x)\ Ca^{2+} + 2xMg^{2+} + NO_3^- + Cl^- + H_2O + 2HCO_3^-$ |
| 6. $(NH_4)_2CO_3$ | $(NH_4)_2CO_3 \rightarrow 2NH_3 \uparrow + H_2O + CO_2 \uparrow$<br>$3Ca_{(1-x)}Mg_xCO_3 + (NH_4)_2CO_3 + 4O_2 \rightarrow 3(1-x)\ Ca^{2+} + 3xMg^{2+} + 2NO_3^- + 2H_2O + 4HCO_3^-$ |
| 7. $Ca_3(PO_4)_2$ | (1) $(1-x)\ Ca^{2+} + xMg^{2+} + 2HCO_3^- \rightarrow Ca_{(1-x)}Mg_xCO_3 + CO_2 + H_2O$<br>(2) $Ca + PO_4 \rightarrow Ca\text{-}P$ |
| 8. $(NH_4)_3PO_4$ | (1) $2Ca_{(1-x)}Mg_xCO_3 + NH_4^+ + 2O_2 \rightarrow 2(1-x)\ Ca^{2+} + 2xMg^{2+} + NO_3^- + H_2O + 2HCO_3^-$<br>(2) $Ca + PO_4 \rightarrow Ca\text{-}P$ |
| 9. Ca-Mg-P | (1) $(1-x)\ Ca^{2+} + xMg^{2+} + 2HCO_3^- \rightarrow Ca_{(1-x)}Mg_xCO_3 + CO_2 + H_2O$<br>(2) $Ca + PO_4 \rightarrow Ca\text{-}P$ |
| 10. Urea | $3Ca_{(1-x)}Mg_xCO_3 + CO(NH_2)_2 + 4O_2 \rightarrow 3(1-x)\ Ca^{2+} + 3xMg^{2+} + 2NO_3^- + 4HCO_3^-$ |
| 11. $K_2CO_3$ | (i) $(1-x)\ Ca^{2+} + xMg^{2+} + 2HCO_3^- \rightarrow Ca_{(1-x)}Mg_xCO_3 + CO_2 + H_2O$<br>(ii) $K_2CO_3 + H_2O \rightarrow 2K^+ + HCO_3^- + OH^-$ |

Note: (1) Common ion effect: The $Ca_{(1-x)}Mg_xCO_3$ produced when the concentrations of $Ca^{2+}$, $Mg^{2+}$ and/or $HCO_3^-$ increases (for treatment 7, 9 and 11): $(1-x)\ Ca^{2+} + xMg^{2+} + 2HCO_3^- \rightarrow Ca_{(1-x)}Mg_xCO_3 + CO_2 + H_2O$;

(2) Inhibition of calcite dissolution/precipitation by phosphate: calcium orthophosphate (Ca-P) precipitation produced on the surface of calcite after the addition of $PO_4^{3-}$ in soil, resulting in the inhibition of the dissolution/precipitation of calcite (for treatment 7, 8 and 9): $Ca + PO_4 \rightarrow Ca\text{-}P$

Table 4 The amount of fertilizer-derived $NH_4^+$ at the initial phase of the experiment and the
potential nitrogenous transformation ($NH_4^+$-$NO_3^-$)

| Treatment | Molecular mass g/mol | Amount of added fertilizer /g | Molar amount of added fertilizer /mole | Amount of fertilizer-derived $NH_4^+$ /mole | The maximum of N products /mole |
|---|---|---|---|---|---|
| $NH_4NO_3$ | 80 | 43 | 0.54 | 0.54 | 1.08 |
| $NH_4HCO_3$ | 79 | 85 | 1.08 | 1.08 | 1.08 |
| $NaNO_3$ | 85 | 91 | 1.07 | 0.00 | 1.07 |
| $NH_4Cl$ | 53.5 | 57 | 1.07 | 1.07 | 1.07 |
| $(NH_4)_2CO_3$ | 96 | 51 | 0.53 | 1.06 | 1.06 |
| $Ca_3(PO_4)_2$ | 310 | 52 | 0.17 | 0.00 | 0.00 |
| $(NH_4)_3PO_4$ | 149 | 15 | 0.10 | 0.30 | 0.30 |
| Ca-Mg-P | nd | 44 | nd | 0.00 | 0.00 |
| Urea | 60 | 32 | 0.53 | 1.06 | 1.06 |
| $K_2CO_3$ | 138 | 10 | 0.07 | 0.00 | 0.00 |

nd=no data. The amount of added fertilizer (g) divided by its molecular mass (g/mol) is the molar
amount of fertilizer (mole). The amounts of fertilizer-derived $NH_4^+$ are calculated by their own
ionization or hydrolysis processes. The maximum of N products is estimated by their main
reactions in Table 3.

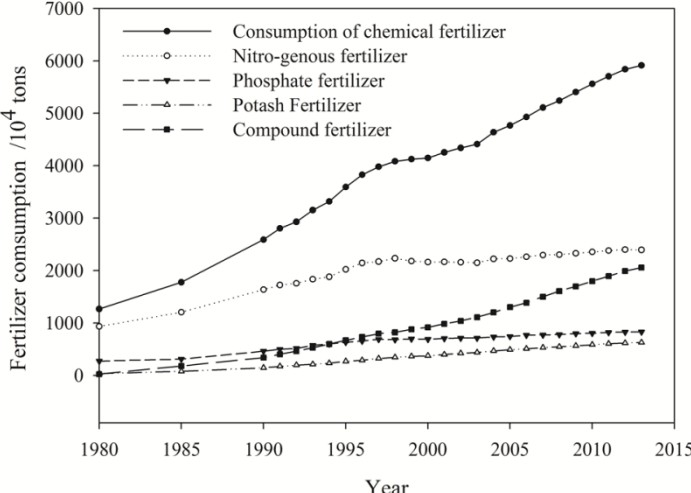


Fig. 1 The change in chemical fertilizer consumption in China during the 1980-2013 period

The data were collected from National Bureau of Statistics of the People's Republic of China

(NBS, 2014) (http://www.stats.gov.cn/tjsj/ndsj/)



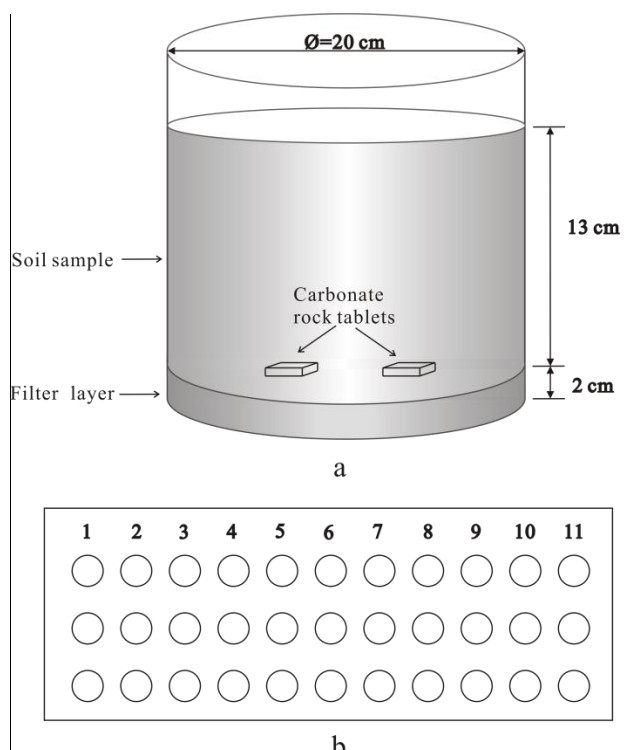

a

b



Fig. 2 Sketch of the soil column (a) and their on-site layout (b)

Fig. 2a: The filter layer (2 cm thick) consists of gravel, coarse sand and fine sand; Fig. 2b: 11

fertilization treatments with 3 replicates including: 1. Control; 2. $NH_4NO_3$; 3. $NH_4HCO_3$; 4. $NaNO_3$; 5.

$NH_4Cl$; 6. $(NH_4)_2CO_3$; 7. $Ca_3(PO_4)_2$; 8. $(NH_4)_3PO_4$; 9. Ca-Mg-P; 10. Urea; 11. $K_2CO_3$.





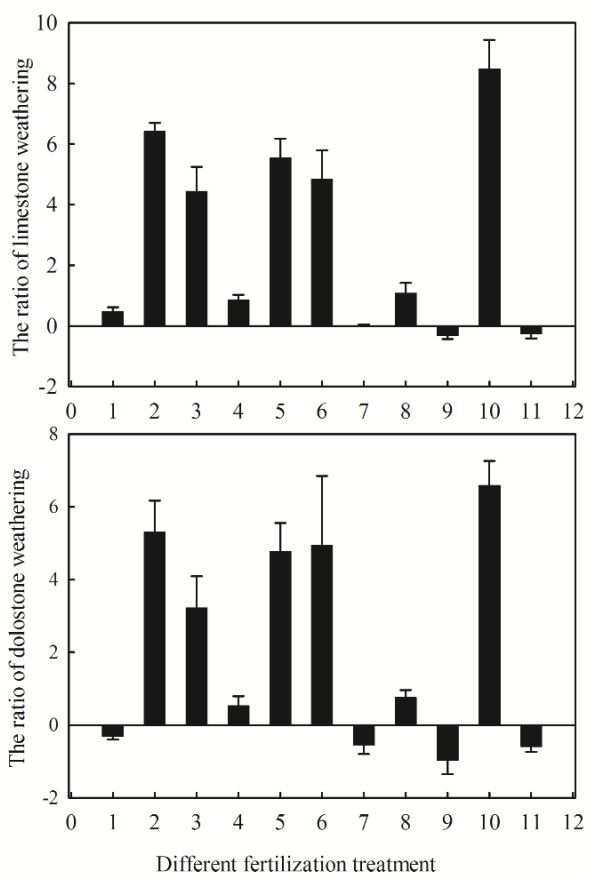


Fig. 3 The R$w$ (‰) of limestone and dolostone under different fertilizer treatments

Treatment 1. Control; 2. $NH_4NO_3$; 3. $NH_4HCO_3$; 4. $NaNO_3$; 5. $NH_4Cl$; 6. $(NH_4)_2CO_3$; 7.

$Ca_3(PO_4)_2$; 8. $(NH_4)_3PO_4$; 9. Ca-Mg-P; 10. Urea; 11. $K_2CO_3$. R$w$ = 1000(W$i$-W$f$)/W$i$, where W$i$ is
the initial weight of the carbonate rock tablet, and W$f$ is the final weight.

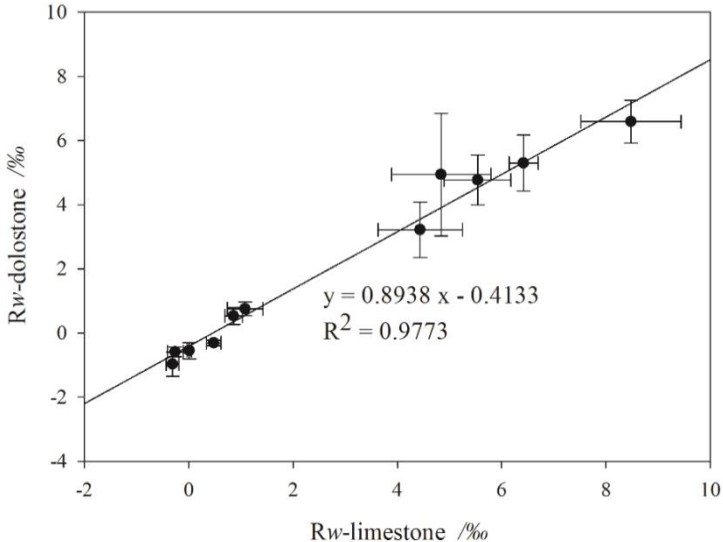


Fig. 4 The linear correlation of R$w$ (‰) of limestone and dolostone

R$w$ = 1000(W$i$-W$f$)/W$i$, where W$i$ is the initial weight of the limestone tablet, and W$f$ is the final

weight.


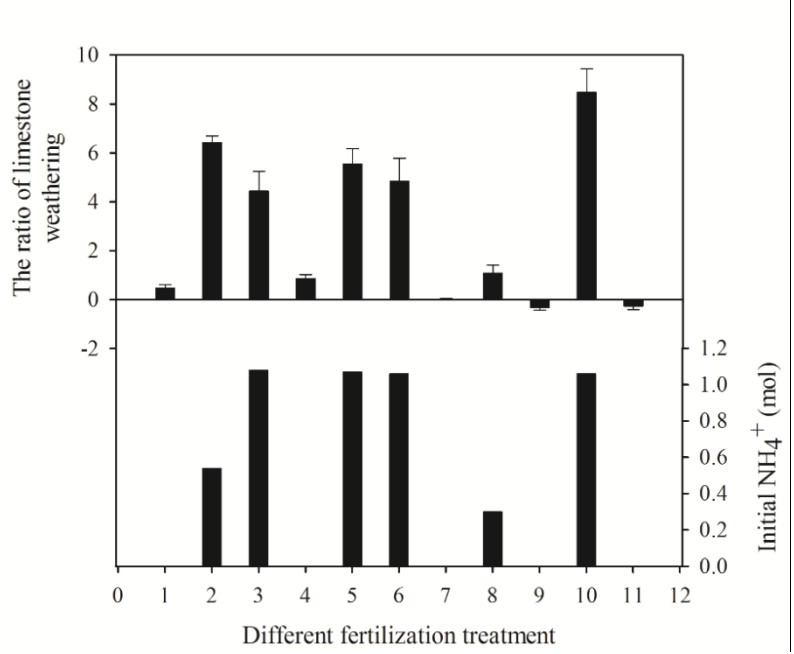

Fig. 5 The R*w* (‰) of limestone and the molar amount of produced $NH_4^+$ under different fertilizer treatments

Treatment 1. Control; 2. $NH_4NO_3$; 3. $NH_4HCO_3$; 4. $NaNO_3$; 5. $NH_4Cl$; 6. $(NH_4)_2CO_3$; 7. $Ca_3(PO_4)_2$; 8. $(NH_4)_3PO_4$; 9. Ca-Mg-P; 10. Urea; 11. $K_2CO_3$. R*w* =1000(W*i*-W*f*)/W*i*, where W*i* is the initial weight of the limestone tablet, and W*f* is the final weight.