# Peer review of "Impact of different fertilizers on the carbonate weathering in a typical karst area,"

_Earth Surface Dynamics, 2016_

## Referee Comment (RC1) · Anonymous Referee #1 · 2 Nov 2016

This study presents the responses of carbonate weathering to different fertilizer addition using the field column experiment. In this study, the authors analyzed weathering amount and ratio of carbonate rock tablets after one year field experiment. Especially, it is interesting that authors identified the variability of carbonate weathering rate under various treatment. The data interpretation and organization in this manuscript is clear. This study provides valuable data and good measurement of carbonate weathering. However, the manuscript still has some points need to be further clarified. 1. Could authors provide the ratio of the nitrate fertilizer vs. total nitrogen fertilizers in studied area, or China, or whole world? It is important for the significance of the manuscript and the experiment. 2. Could authors compare results with data by Prof. Yuan DX's group

[Figure]

? 3. The manuscript need more detail for the experiment. L134-L139: Please give a detail introduction for the added amount of fertilizers in these treatments. It seems that the added amount of nitrogen is slight difference. What's the proportion of these eleven fertilization treatments in local practical use? Why choose the added amount of fertilizers are 30 times than its local practical amount? The application fertilizers in local practical may change two or three times to use. Do you think the added fertilizers by one time may affect the result? Why don't authors set different height for this experiment, which might be more interesting? Does author consider that the land use influences the carbonate weathering in the experiment? 4. The authors made three replications. So please show the data errors for each average value. 5. Could you assess the variation of nitrate fertilizer change in the column. Then understand the balance between acid producing and carbonate weathering together. 6. Line 15-17, the sentence has too many "different"s. Please revise it. 7. In section 2.2, could authors provide details about abbreviation of OM, ASI method and others when you write that at the first time. Please check that in whole manuscript. 8. L162-166 It seems Table 2 and Fig3 are repeated. 9. L182-L197 This paragraph can be removed to the introduction. 10. L213-L219 It is repeated the introduction (L49-L54). 11. Major conclusion might be revised. Ammonium fertilizer mainly includes $NH_4NO_3$, $NH_4Cl$, $(NH_4)_2CO_3$ fertilizers, not includes urea fertilizer. I suggest reductive nitrogenous fertilizer could enhance carbonate weathering via nitrification.

---

## Referee Comment (RC2) · Anonymous Referee #2 · 5 Dec 2016

General comments

This paper is dealing with the role of the application of chemical and organic fertilizers on carbonate weathering in a karst area. Their approach was based on a laboratory experiment using a soil column including two carbonate rock tablets over one year on the field. The authors discussed the loss or gain of weight of each rock tablets in term of variability of carbonate weathering under various fertilizer treatments. The topic as such should be well suited for a publication in Earth Surface Dynamics, but this manuscript is not, at least in its present state. It needs some clarifications. I would suggest major revisions of the present manuscript.

Specific comments In its present form, I cannot recommend the publication of this

manuscript for different reasons.

1 - The authors did not present very well the process/method of weathering which has been used in this experiment: did the authors perform a leaching of the soil column? How are the fertilizers introduced in the soil column? Are spread mixed with soil or spread in solutions? The lack of explanation of the method used does not allow us to assess the results at their fair value. There is also a lack of discussion and comparison of numerical values obtained in other experiments and in natural and agricultural catchments. The carbonate weathering is only estimated based on the weight of each rock tablets. It is not checked by the geochemistry of both rock tablets and the potential weathering/soil solution. Indeed, it would have been interesting to have an estimation of the chemical weathering. 2 – To speed up the carbonate weathering, the fertilizers were introduced by increasing their amount by 30 times (Why 30 times?). It is a bit problematic, because the authors changed the soil/fertilizers ratio compared to "natural/anthropogenic" ratio? What is this ratio in the local agricultural catchments? What are the specificities these local catchments compared to national Chinese catchments and worldwide catchments? 3 – The variability of the experimental replicates should be shown (average and standard deviations), presented and discussed. This can be presented in Table 2. 4 – In general, the authors used limestone and dolostone tablets. They did not discuss the results of dolostone tablets, only those from limestone tablets. In the discussion, the difference or similarity between dolostone and limestone is erased as the authors discuss about carbonates. More attention, or at least an explanation about the use of the general term of "carbonates" instead of the difference between dolostone and limestone should be given.

Technical corrections

Here are some specific comments and suggestions. ‐ In several times in the manuscript (last sentence of the abstract, first paragraph of the results, 4.4. and the last sentence of the conclusion) the authors used the expression "can aid carbonate weathering": they should precise if the fertilizers enhance, increase, or decrease carbonate weathering...

‐ Introduction: - L.43 - The authors should add references showing the relationship between carbonate weathering and climate in addition to Liu et al. (2010, 2011); for example Kump et al., 2000). - L.47 - The authors should precise that the disturbance of CO2 consumption disturbance may be overestimated at a local scale by taking into account Ca2+ and Mg2+ produced by a natural carbonate weathering and those produced indirectly by anthropogenic activities in the watershed. And what about this disturbance at a global scale?

‐ 2.2. Soil properties : - At which depth did the authors sample their soils? - Should precise pH(H2O) - Precise what OM means: organic matter I suppose. - Precise what ASI method means. - What is the soil typology?

‐ 2.3. Soil column - What is the filter material? - What kind of carbonate rocks did the authors use for their experiment? Are they reference rocks or rocks from karst area of HuaXi district? - How did the authors deposit each fertilizer in the column? In liquid or solid form? - At which temperature has the experiment been performed? - Did you leach the soil column with a solution? If yes, with which solution? - In figure 2: the authors draw 3 rock tablets, while the authors put only 2 rock tablets at the bottom of the column. Should change it. - Did the authors perform the same experiment without rock tablets if they leach their column in order to observe the leaching solution of the column? - Did the authors put the 2 different rock tablets (calcite and dolomite) in the same column? - The authors should explain the reason of the fertilizer weight use in the experiment.

‐ 3. Results - L.164-165: Do not repeat Table 2 and Fig. 3. You may write: "The results are presented in Table 2 and in Figure 3."

‐ 4. Discussion - 4.1.: the first paragraph (L. 182-197) is quite general and it would be worthy to move it either in the introduction, or at least in the Materials and Methods section. - 4.1. L.213-219: It is exactly the same text as in the introduction (L. 48-54).

The authors may express their idea at least a little bit differently. - Information about soils and soil solutions are needed in order to understand their chemical evolution during the carbonate weathering. - Would it be possible to present the chemistry of each fertilizer used in this experiment? This can be added in supplementary information.
* * *

---

## Author Comment (AC1) · 3 Jan 2017

Referee 1 Comment 1: Could authors provide the ratio of the nitrate fertilizer vs. total nitrogen fertilizers in studied area, or China, or whole world? It is important for the significance of the manuscript and the experiment. Answer: Yes, it is very important, but we can't find the specific published data about it. Only one figure we can calculate according to a reference published in 2008 is that the global production of NH4NO3 accounts for 10% of the total N fertilizers, and about 4% in China. But we think this figure is not fit to be cited in this manuscript. We think that this study only focused on the relative potential mechanism, and the study on the estimation of the impacts will be considered in the future. Comment 2: Could authors compare results with data by

Prof. Yuan DX's group paper ? Answer: We have cited the relative result from Prof. Yuan Dx's group in this study, such as the papers from Jiang Z., et al.; Liu Z., et al. and Jiang Y., et al.. In fact, we are familiar with them and their study focuses, we know that their studies are a little bit different from ours so far. Comment 3: The manuscript need more detail for the experiment. L134-L139: Please give a detail introduction for the added amount of fertilizers in these treatments. It seems that the added amount of nitrogen is slight difference. (1) What's the proportion of these eleven fertilization treatments in local practical use? Answer: The added amounts of these 11 fertilizers were designed only by the average amount of N, P and K fertilizer in the local practical use. Changed in the manuscript: We have added the amount of N, P and K fertilizer in local practical use in this manuscript like this: N fertilizer: 160 kg N• ha-1; P fertilizer: 150 kg P2O5• ha-1; K fertilizer: 50 kg K• ha-1) (2) Why choose the added amount of fertilizers are 30 times than its local practical amount? The application fertilizers in local practical may change two or three times to use. Do you think the added fertilizers by one time may affect the result? Answer: Because the added amount of fertilizer can magnify and quicken the fertilization effect in the short-term according to another experiment from us, and can't affect the phenomenon we want to observe in this study. Another paper of ours (in preparation) about a series of different amount of fertilizer addition will discuss this issue. (3)Why don't authors set different height for this experiment, which might be more interesting? Answer: Thank you for your good suggestion. This study is conducted as a simply start. The suggestion on different height must be interesting. We are considering that in further study. (4)Does author consider that the land use influences the carbonate weathering in the experiment? Answer: In fact, there are some good studies to be published about that around the world. But most of them are conducted by the search and evaluation of riverine hydro-geochemical data. Because of this, we did this study from another angle, and hope to connect them in the future. Comment 4: The authors made three replications. So please show the data errors for each average value. Answer: Fig.3 has showed these error bars. We will add them in Table 2. Changed in the manuscript: The error data have been added in

Table 2. Comment 5: Could you assess the variation of nitrate fertilizer change in the column. Then understand the balance between acid producing and carbonate weathering together. Answer: We tried our best to explain the chemical process of N fertilizer in section 4.3 with relative chemical reactions. Comment 6: Line 15-17, the sentence has too many "different"s. Please revise it. Changed in the manuscript: We use "these discrepancies" instead of "their differences". Comment 7: In section 2.2, could authors provide details about abbreviation of OM, ASI method and others when you write that at the first time. Please check that in whole manuscript. Changed in the manuscript: We have changed this. Comment 8: L162-166 It seems Table 2 and Fig3 are repeated. Changed in the manuscript: We deleted the relative data in Table 2, and leave the Fig. 3 because the figure is easier to see. And we also changed corresponding texts. Comment 9: L182-L197 This paragraph can be removed to the introduction. Changed in the manuscript: We moved them to the introduction. Comment 10: L213-L219 It is repeated the introduction (L49-L54). Answer: Because they are for elaborating different problem, we think we should put one of them another way. Changed in the manuscript: We changed the statements in section 4.2. Comment 11: Major conclusion might be revised. Ammonium fertilizer mainly includes NH4NO3, NH4Cl, (NH4)2CO3 fertilizers, not includes urea fertilizer. I suggest reductive nitrogenous fertilizer could enhance carbonate weathering via nitrification. Changed in the manuscript: Yes, the statement that nitrogenous fertilizer can aid carbonate weathering should be replaced by ammonium fertilizer" in this manuscript is not precise. We deleted it.

---

## Author Comment (AC2) · 3 Jan 2017

Referee 2 Comment 1 - The authors did not present very well the process/method of weathering which has been used in this experiment: (1)did the authors perform a leaching of the soil column? How are the fertilizers introduced in the soil column? Are spread mixed with soil or spread in solutions? The lack of explanation of the method used does not allow us to assess the results at their fair value. There is also a lack of discussion and comparison of numerical values obtained in other experiments and in natural and agricultural catchments. The carbonate weathering is only estimated based on the weight of each rock tablets. It is not checked by the geochemistry of both rock tablets and the potential weathering/soil solution. Indeed, it would have been

interesting to have an estimation of the chemical weathering. Answer: The fertilizer was mixed with soil before filling in columns. Changed in the manuscript: We added a sentence to explain this. The soil was weighed, mixed perfectly with above fertilizer, respectively, and filled in its own column. Comment 2 – To speed up the carbonate weathering, the fertilizers were introduced by increasing their amount by 30 times (Why 30 times?). It is a bit problematic, because the authors changed the soil/fertilizers ratio compared to "natural/anthropogenic" ratio? What is this ratio in the local agricultural catchments? What are the specificities these local catchments compared to national Chinese catchments and worldwide catchments? Answer: Because the added amount of fertilizer can magnify and quicken the fertilization effect in the short-term according to another experiment from us, and can't affect the phenomenon we want to observe in this study. Another paper of ours (in preparation) about a series of different amount of fertilizer addition will discuss this issue. The added amounts of these 11 fertilizers were designed only by the average amount of N, P and K fertilizer in the local practical use. Changed in the manuscript: We have added the amount of N, P and K fertilizer in local practical use in this manuscript like this: (N fertilizer: 160 kg N• ha-1; P fertilizer: 150 kg P2O5• ha-1; K fertilizer: 50 kg K• ha-1) Comment 3 – The variability of the experimental replicates should be shown (average and standard deviations), presented and discussed. This can be presented in Table 2. Answer: We did it. Comment 4 – In general, the authors used limestone and dolostone tablets. They did not discuss the results of dolostone tablets, only those from limestone tablets. In the discussion, the difference or similarity between dolostone and limestone is erased as the authors discuss about carbonates. More attention, or at least an explanation about the use of the general term of "carbonates" instead of the difference between dolostone and limestone should be given. Answer: The difference between limestone and dolostone is not noteworthy, so we use carbonate instead. Yes, we need to give some sentences to explain this. Changed in the manuscript: We added the statement "The result between limestone and dolostone weathering under different fertilization treatment were similar. We will explain the results with carbonates instead of individual

dolostone and limestone." in this manuscript to explain Comment 5: In several times in the manuscript (last sentence of the abstract, first paragraph of the results, and the last sentence of the conclusion) the authors used the expression "can aid carbonate weathering": they should precise if the fertilizers enhance, increase, or decrease carbonate weathering. Changed in the manuscript: The statement that nitrogenous fertilizer can aid carbonate weathering should be replaced by ammonium fertilizer" in this manuscript is not precise. We deleted it. And we replaced the rest aids with the word "increase". Comment 6: Introduction: - L.43 - The authors should add references showing the relationship between carbonate weathering and climate in addition to Liu et al. (2010, 2011); for example Kump et al., 2000). – Changed in the manuscript: We added it. Comment 7: L.47 - The authors should precise that the disturbance of CO2 consumption disturbance may be overestimated at a local scale by taking into account Ca2+ and Mg2+ produced by a natural carbonate weathering and those produced indirectly by anthropogenic activities in the watershed. And what about this disturbance at a global scale? Answer: Here, we are just trying to introduce the potential disturbance at the regional/global scales by summarizing and classifying some references in the 1st paragraph. And the specific disturbances from fertilizer addition were further discussed in the 2nd paragraph. Comment 8: 2.2. Soil properties : - At which depth did the authors sample their soils? - Should precise pH(H2O) - Precise what OM means: organic matter I suppose. - Precise what ASI method means. - What is the soil typology? Answer: The pH had been listed in Table 1. Changed in the manuscript: The meanings of OM and ASI have been added. We changed the statement "The soil used in this column experiment was sampled from the B horizon (below 20 cm in depth) of yellow-brown soil in a cabbage-corn or capsicum-corn rotation plantation in Huaxi district." to explain the soil samples and typology. Comment 9: 2.3. Soil column - What is the filter material? Answer: Yes, it is a misleading expression here. Changed in the manuscript: It has been changed into: A Polyethylene net (Ø 0.5 mm) was placed in the bottom of the columns to prevent soil loss. A filter sand layer with 2 cm thickness including gravel, coarse sand and fine sand was spread on the net. Comment 10: What kind

**ESurfD**
of carbonate rocks did the authors use for their experiment? Are they reference rocks or rocks from karst area of HuaXi district? Answer: yes, it was collected from karst area of Huaxi district. Changed in the manuscript: We added this information in this manuscript. Comment 11: How did the authors deposit each fertilizer in the column? In liquid or solid form? At which temperature has the experiment been performed? - Did you leach the soil column with a solution? If yes, with which solution? Answer: The soil fertilizer was weighed and mixed with soil before filling in columns. Changed in the manuscript: We added a sentence to explain this. The soil was weighed, mixed perfectly with above fertilizer, respectively, and filled in its own column. Comment 12: - In figure 2: the authors draw 3 rock tablets, while the authors put only 2 rock tablets at the bottom of the column. Should change it. Changed in the manuscript: We have changed this. Comment 13: - Did the authors perform the same experiment without rock tablets if they leach their column in order to observe the leaching solution of the column? Answer: We didn't design that in this study. We didn't collect the soil solution. The leaching depended on the rainfall. Comment 14:- Did the authors put the 2 different rock tablets (calcite and dolomite) in the same column? Answer: Yes, we did. Comment 15: The authors should explain the reason of the fertilizer weight use in the experiment. Answer: Because the added amount of fertilizer can magnify and quicken the fertilization effect in the short-term according to another experiment from us, and can't affect the phenomenon we want to observe in this study. Another paper of ours (in preparation) about a series of different amount of fertilizer addition will discuss this issue. The added amounts of these 11 fertilizers were designed only by the average amount of N, P and K fertilizer in the local practical use. Changed in the manuscript: We have added the amount of N, P and K fertilizer in local practical use in this manuscript like this: N fertilizer: 160 kg N  ha-1; P fertilizer: 150 kg P2O5  ha-1; K fertilizer: 50 kg K  ha-1) Comment 16: 3. Results - L.164-165: Do not repeat Table 2 and Fig. 3. You may write: "The results are presented in Table 2 and in Figure 3. Changed in the manuscript: We have changed this. Comment 17: 4. Discussion - 4.1.: the first paragraph (L. 182-197) is quite general and it would be worthy to move it

either in the introduction, or at least in the Materials and Methods section. Changed in the manuscript: We moved them to the introduction. Comment 18: 4.1. L.213-219: It is exactly the same text as in the introduction (L. 48-54) The authors may express their idea at least a little bit differently. Answer: Because they are for elaborating different problem, we think we should put one of them another way. Changed in the manuscript: We changed the statements in section 4.2. Comment 19: Information about soils and soil solutions are needed in order to understand their chemical evolution during the carbonate weathering. - Would it be possible to present the chemistry of each fertilizer used in this experiment? This can be added in supplementary information. Answer: yes, it is very important. Most of them have been discussed in section 4.2 and 4.3 so far. And we are doing some further research on that.

---

## Editor Comment (EC1) · G. Govers (Editor) · 1 May 2017

Dear authors,

I have now read the comments of both reviewers and have thoroughly read the manuscript myself. My conclusion is that, while the manuscript has been improved, it still needs considerable work. The main points are:

- The English is not up to standard: you really need to make sure the paper is being read/corrected by a native English speaker (or somebody with substantial experience in publishing in English-speaking journals) who is familiar with your domain. - While there is now a statistical analysis, it is not correct (in my opinion). You need a two way ANOVA

as you have both a treatment (fertilizer) and a rock type (dolostone or limestone) effect. You cannot evaluate the treatment effect correctly if you do not account for the rock type effect and vice versa. This needs to be corrected. - The analysis and discussion of the results needs further improvement in presentation and in analysis. You did add some additional data but: - you mention that there are several studies that already studi"ed the effect of fertilizers on carbonate weathering. Yet you do not quantitatively compare your results with those of these studies - in the comparison you make you use a weathering rate expressed in g m\*\*-2 y\*\*-2. This raises several questions: - Grams of what ? CaCO3 or Ca ? Or rock ? - How do you convert the results of your tablet experiments to a rate per unit of surface area; as far as I can see this is nowhere explained in the text. - The presentation needs also other improvements: Table 3 is a case in point. You should provide all weathering reactions that a reader needs to understand the calculations you make (e.g. about the amount of NH4 released per mole of fertilizer). This needs to be done in consistent way: give the chemical formulation of every fertilizer, make sure that numbers are given to the remarks only.

- There are also substantial remarks in the report of reviewer 2 that you need to address. Please read them carefully and respond to all of them.

- You will find more remarks in the manuscrpt file that is attached. Please read them carefully and respond to all of them.

I invite you to resubmit your manuscript after thorough revision and I will send it out again to reviewers but only if I am convinced that the level of English and the organisation of the manuscript is as can be expected for an E-Surf manuscript. If this is not the case I will return the manuscript to you and you will be free to submit it elsewhere.

With very best wishes, Gerard Govers

Please also note the supplement to this comment:
http://www.earth-surf-dynam-discuss.net/esurf-2016-50/esurf-2016-50-EC1-

supplement.pdf

---

## Author Comment (AC3) · 10 Jun 2017

1- The English is not up to standard: you really need to make sure the paper is being read/corrected by a native English speaker (or somebody with substantial experience in publishing in English-speaking journals) who is familiar with your domain.

Changed in the manuscript: we have re-edited the language question by the Language Services of Elsevier.

2. While there is now a statistical analysis, it is not correct (in my opinion). You need a two way ANOVA as you have both a treatment (fertilizer) and a rock type (dolostone

or limestone) effect. You cannot evaluate the treatment effect correctly if you do not account for the rock type effect and vice versa. This needs to be corrected.

Responses: Thanks for your pointing it out. Given that the primary objective of the statistical analysis is to test the accordance between limestone and dolostone, we ploted the linear correlation diagram of the dissolution rates for these two rocks to make this problem clearer and easy to understand. Changed in the manuscript: we added Fig. 4, a linear correlation diagram ($R^2$=0.9773), instead, to illustrate no difference in the dissolution rates between limestone and limestone. See the details in the manuscript.

3.- The analysis and discussion of the results needs further improvement in presentation and in analysis. You did add some additional data but: - you mention that there are several studies that already studied the effect of fertilizers on carbonate weathering. Yet you do not quantitatively compare your results with those of these studies Responses: we had done some comparison our results with others in section 4.4. The conclusion is that it is difficult to compare between the results from the carbonate-rock-tablet test and the riverine hydro-chemical method.

4. In the comparison you make you use a weathering rate expressed in g m-2 y-1. This raises several questions: -Grams of what ? $CaCO_3$ or Ca ? Or rock ? - How do you convert the results of your tablet experiments to a rate per unit of surface area; as far as I can see this is nowhere explained in the text. Responses: In fact, we have mentioned it in Methods section. Raw = (Wi-Wf)/(S*T) (3) where Wi is the initial weight of the carbonate rock tablets, Wf is their final weights, S is the surface area of carbonate rock tablets, and T is the length of the experimental period. Wi and Wf is grams of carbonate rocks (limestone or dolostone) that we used in this study. S is the surface area of rock tablets. Changed in the manuscript: in order to make it clearer, we added an information in Table 2: Raw = (Wi-Wf)/(S*T), where Wi is the initial weight of the carbonate rock tablets, and Wf is their final weight. S is the surface area of carbonate rock tablets (In this study, we used a same S=7 cm2 for every tablets), and T is the experiment period.

5.- The presentation needs also other improvements: Table 3 is a case in point. You should provide all weathering reactions that a reader needs to understand the calculations you make (e.g. about the amount of NH4 released per mole of fertilizer). This needs to be done in consistent way: give the chemical formulation of every fertilizer, make sure that numbers are given to the remarks only. Responses: Some treatments are difficult to describe with a specific reaction, E.g.: for treatment 7, 8, 9, 11 etc. we used a simply reaction on inhibition of phosphate to calcite dissolution/precipitation (Ca + PO4 → Ca-P) instead. Changed in the manuscript: we polished the Table 3. See it in the manuscript. We added notes in table 3: (1) Common ion effect: The Ca(1-x)MgxCO3 produces when the concentrations of Ca2+ Mg2+ and/or HCO3- increases (for Treatment 7, 9 and 11): (1-x) Ca2+ + xMg2+ + 2HCO3-→ Ca(1-x)MgxCO3 + CO2 + H2O (2) Inhibition of phosphate to calcite dissolution/precipiation: calcium orthophosphate (Ca-P) precipitation produces on the surface of calcite after the addition of PO43- in soil, resulting in inhibiting the dissolution/precipitation of calcite (for Treatment 7, 8 and 9): Ca + PO4 → Ca-P We added notes in table 4: wd=without data; The amount of added fertilizer (g) divided by its molecular mass (g/mol) was the molar amount of fertilizer (mole); The amounts of fertilizer-derived NH4+ is calculated by their own ionization or hydrolysis processes. The maximum of N products is estimated by their main reactions in table 3.

6- There are also substantial remarks in the report of reviewer 2 that you need to address. Please read them carefully and respond to all of them. - You will find more remarks in the manuscript file that is attached. Please read them carefully and respond to all of them. Responses: In fact, we had revised according to the reviewer 2 including remarks in PDF file. Here, we give a point to point response and revision description.   Responses and revision descriptions on reviewer 2' comments in pdf manuscript

Abstract 1. Note in "fertilization": You should add a sentence after this, stating as to why fertilization may affect carbonate weathering (release of protons...) Changed in the manuscript: we added the clause to interpret. "since the addition of fertilizers tends to change the chemical characteristics of soil such as pH value".

2. Revision suggestions about English language in the rest part of Abstract section Changed in the manuscript: we changed it according to these suggestions

Introduction 3. The statement of the sentence " However, a disturbance to CO2........ of N-fertilizer. Changed in the manuscript: It is changed into: However, fluvial alkalinity may also be produced by other processes including the reaction between carbonates and the protons derived (i) from the nitrification of N-fertilizer (Barnes and Raymond, 2009; Chao et al., 2011; Gandois et al., 2011; Hamilton et al., 2007; Oh and Raymond, 2006; Perrin et al., 2008; Pierson-wickmann et al., 2009; Semhi and Suchet, 2000; West and McBride, 2005), (ii) from the sulfuric acid (Lerman and Wu, 2006; Lerman et al., 2007; Li et al., 2008; Li et al., 2009), (iii) from organic acid secreted by microorganisms (Lian et al., 2008) as well as (iv) from acidic soil (Chao et al., 2014ïïjŻ 2017).

4. Note in "acidic soil": Can you be more specific about the process here ? What do you mean here ? Responses: The acidity or proton in acidic soil can lead to carbonate weathering. Changed in the manuscript: we changed the statement and added a new reference ïïjĹChao, S., et al., Impact of animal manure addition on the weathering of agricultural lime in acidic soils: The agent of carbonate weathering. Journal of Groundwater Science and Engineering, 2017. 5(2): p. 202-212.ïïjĽ

5. Note in "the deficit of CO2 untake due to N-fertilizer addition: This is a strange formulation, difficult to understand what you really mean: is this not 'estimated that N-fertilizers contributed up to Changed in the manuscript: we changed it into: Perrin et al. (2008) estimated that the contribution of N-fertilizer (usually in form of NH4NO3) represent up to 5.7-13.4% and 1.6-3.8% to carbonate dissolution for France and on a global scale, respectively.

6. Note in "Our results show that..... experiment": You cannot discuss your results before you presented them. Changed in the manuscript: we changed it into: yet it is a preferred option for the condition controlled contrast or stimulated experiment (Chao et al., 2017; Chao et al., 2014; Chao et al., 2011). 7. Revision suggestions about English language in the rest part of Introduction section Changed in the manuscript: we changed it according to these suggestions.

Materials and Methods 8. Note in "2013"(section 2.1): Also give the surface area of the agricultural area in the district so that the reader knows what the average application rate per unit of surface area is. Changed in the manuscript: we changed "0.8 Mt in 1980 to 1.0 Mt in 2013" into "from 150 kg/ha in 1980 to 190 kg/ha in 2013".

9. Note in " the amount of added fertilizers" (section 2.3): How was the fertilizer applied ? Changed in the manuscript: we added the information to explain this point. The 6 kg soil was weighed (bulk density=1.3 g/cm3), mixed perfectly with above fertilizer, respectively, and filled in its own column

10. Note in "3-5%" ( section 2.4) "The percentages do not add up: if there is min 98% crysal dolomite you cannot have 3% calcite". Changed in the manuscript: we changed it into: dolostone with 98-99% power crystal dolomite, 1% pyrite and trace quantities organic matter.

11. Revision suggestions about English language in the rest part of Materials and Methods section Changed in the manuscript: we changed it according to these suggestions.

Results 12. Note in "carbonate" ( section 3) : So, you finally calculated carbonate loss: then this should be defined higher up. Changed in the manuscript: we note this point, we re-organized this section.

13. Note in "Acw, Rcw and Racw" (section 3): Is this distinction between rates and amounts meaningful. I do not think so as the experiments were carried out (as far as I understand) over a single, fixed time period. Responses: we noted this problem. Changed in the manuscript: Considering this point, we deleted some of them, and

re-wrote this paragraph.

14. Note in "-0.0028g and -0.0007g for limestone and dolomite" (section 3): You mention in the methods section that you will discuss results in terms of carbonate loss: this is not what you do here. You need to be consistent. Changed in the manuscript: we deleted them.

15. Note in "of two different carbonate in. . . . . .. a negative value" (section 3): I do not understand this. Responses: it can be easily understood according to the formula of Rw and Raw. Rw = (Wi-Wf)/ Wi Raw = (Wi-Wf)/(S*T) where Wi is the initial weight of the carbonate rock tablets, Wf is their final weights, S is the surface area of carbonate weathering tablets, and T is the length of the experimental period. If the Rw and Raw is positive, it shows carbonate dissolves; if negative, carbonate mineral produces.

16. Revision suggestions about English language in the rest part of Results section Changed in the manuscript: we changed it according to these suggestions.

Discussion 17. Note in "in this study, . . .. . . are therefore valid and credible" (section 4): The fact that the columns were the same is logical: to what extent the results are 'valid' is determined your results, not your judgment. Responses: we noted this problem. Changed in the manuscript: Considering this point, we deleted them

18. Note in "which can originate from . . .. . .acidic soil (Chao et al. 2014)" (section 4): This is a repetition of what has already been said in the text Responses: it is used for interpreting the different problem Changed in the manuscript: we changed it into another one to avoid repeating.

19. Note in "in habiting that the. . .. . . fertilizer amendment" (section 4): unclear Changed in the manuscript: we deleted it.

20. Note in "The Rcw of limestone tablets. . .. .. Fig. 4" (section 4): This can be much clearer presented Note in "results" (section 4): I do not clearly see how you can have exactly 1 mole of NH4: instead you should calculated the amount of fertilizer-derived

NH4 per unit surface area of your columns. Changed in the manuscript: we added much information to make it clearer, including Figures and Tables.

21. Note in page 11(section 4): There also important differences between the treatments which you do not discuss. The weathering rate under treatment 3 is only half of that of treatment 10 (urea) Changed in the manuscript: we added much information to interpret it. We have noted that the Rw values in NH4HCO3 and (NH4)2CO3 treatment are lower than even half of those in urea treatment in spite of adding the same amount of fertilizer-derived NH4 (about 1.07 mole). This is probably because the two fertilizers, NH4HCO3 and (NH4)2CO3, are easier to decompose and produce the NH3 and CO2 gases as following Eq. (20) and (21), resulting in the amount of fertilizer-derived NH4 of lower than 1.07 moles. NH4HCO3 → NH3 ↑ + H2O + CO2 ↑ (20) (NH4)2CO3 → 2NH3 ↑ + H2O + CO2↑ (21)

22. Note in" the enhanced HCO3-"(section 4): Why do we jump to the regional scale here ? Changed in the manuscript: we deleted it.

22. Note in" phosphate "(section 4): Do you have any data supporting this hypothesis ? Responses: it is used for interpreting the different problem

Note in" NaNO3 "(section 4): You give the weathering reactions for all your other treatments, you should also present this one. Perhaps it is better to group all these reactions in a table. Changed in the manuscript: we added the reaction and listed a table including all the reactions.

24. Note in" This will result in doubled overestimation. . . .. carbonate weathering "(section 4): Unclear Responses: For NH4NO3 fertilizer, the (Eq. (12)) show that the two moles of Ca2++Mg2+, NO3- and HCO3- will be produced when one mole NH4NO3 react with 2 moles of carbonate, where only half of NO3- originate from nitrification described as Eq. (8). This will result in a double overestimation on the contribution of the nitrification to carbonate weathering and thus mislead the estimation of CO2 consumption therein.  Responses to Referee 2 Comment 1 - The authors did not present very well the process/method of weathering which has been used in this experiment: (1)did the authors perform a leaching of the soil column? How are the fertilizers introduced in the soil column? Are spread mixed with soil or spread in solutions? The lack of explanation of the method used does not allow us to assess the results at their fair value. There is also a lack of discussion and comparison of numerical values obtained in other experiments and in natural and agricultural catchments. The carbonate weathering is only estimated based on the weight of each rock tablets. It is not checked by the geochemistry of both rock tablets and the potential weathering/soil solution. Indeed, it would have been interesting to have an estimation of the chemical weathering. Answer: The fertilizer was mixed with soil before filling in columns. Changed in the manuscript: We added a sentence to explain this. The soil was weighed, mixed perfectly with above fertilizer, respectively, and filled in its own column. Comment 2 – To speed up the carbonate weathering, the fertilizers were introduced by increasing their amount by 30 times (Why 30 times?). It is a bit problematic, because the authors changed the soil/fertilizers ratio compared to "natural/anthropogenic" ratio? What is this ratio in the local agricultural catchments? What are the specificities these local catchments compared to national Chinese catchments and worldwide catchments? Answer: Because the added amount of fertilizer can magnify and quicken the fertilization effect in the short-term according to another experiment from us, and can't affect the phenomenon we want to observe in this study. Another paper of ours (in preparation) about a series of different amount of fertilizer addition will discuss this issue. The added amounts of these 11 fertilizers were designed only by the average amount of N, P and K fertilizer in the local practical use. Changed in the manuscript: We have added the amount of N, P and K fertilizer in local practical use in this manuscript like this: (N fertilizer: 160 kg NÅů ha-1; P fertilizer: 150 kg P2O5Åů ha-1; K fertilizer: 50 kg KÅů ha-1) Comment 3 – The variability of the experimental replicates should be shown (average and standard deviations), presented and discussed. This can be presented in Table 2. Answer: We did it. Comment 4 – In general, the authors used limestone and dolostone tablets.

They did not discuss the results of dolostone tablets, only those from limestone tablets. In the discussion, the difference or similarity between dolostone and limestone is erased as the authors discuss about carbonates. More attention, or at least an explanation about the use of the general term of "carbonates" instead of the difference between dolostone and limestone should be given. Answer: The difference between limestone and dolostone is not noteworthy, so we use carbonate instead. Yes, we need to give some sentences to explain this. Changed in the manuscript: We added the statement "The result between limestone and dolostone weathering under different fertilization treatment were similar. We will explain the results with carbonates instead of individual dolostone and limestone." in this manuscript to explain Comment 5: In several times in the manuscript (last sentence of the abstract, first paragraph of the results, and the last sentence of the conclusion) the authors used the expression "can aid carbonate weathering": they should precise if the fertilizers enhance, increase, or decrease carbonate weathering. Changed in the manuscript: The statement that nitrogenous fertilizer can aid carbonate weathering should be replaced by ammonium fertilizer" in this manuscript is not precise. We deleted it. And we replaced the rest aids with the word "increase". Comment 6: Introduction: - L.43 - The authors should add references showing the relationship between carbonate weathering and climate in addition to Liu et al. (2010, 2011); for example Kump et al., 2000). – Changed in the manuscript: We added it. Comment 7: L.47 - The authors should precise that the disturbance of CO2 consumption disturbance may be overestimated at a local scale by taking into account Ca2+ and Mg2+ produced by a natural carbonate weathering and those produced indirectly by anthropogenic activities in the watershed. And what about this disturbance at a global scale? Answer: Here, we are just trying to introduce the potential disturbance at the regional/global scales by summarizing and classifying some references in the 1st paragraph. And the specific disturbances from fertilizer addition were further discussed in the 2nd paragraph. Comment 8: 2.2. Soil properties : - At which depth did the authors sample their soils? - Should precise pH(H2O) - Precise what OM means: organic matter I suppose. - Precise what ASI

method means. - What is the soil typology? Answer: The pH had been listed in Table 1. Changed in the manuscript: The meanings of OM and ASI have been added. We changed the statement "The soil used in this column experiment was sampled from the B horizon (below 20 cm in depth) of yellow-brown soil in a cabbage-corn or capsicum-corn rotation plantation in Huaxi district." to explain the soil samples and typology. Comment 9: 2.3. Soil column - What is the filter material? Answer: Yes, it is a misleading expression here. Changed in the manuscript: It has been changed into: A Polyethylene net (Ø 0.5 mm) was placed in the bottom of the columns to prevent soil loss. A filter sand layer with 2 cm thickness including gravel, coarse sand and fine sand was spread on the net. Comment 10: What kind of carbonate rocks did the authors use for their experiment? Are they reference rocks or rocks from karst area of HuaXi district? Answer: yes, it was collected from karst area of Huaxi district. Changed in the manuscript: We added this information in this manuscript. Comment 11: How did the authors deposit each fertilizer in the column? In liquid or solid form? At which temperature has the experiment been performed? - Did you leach the soil column with a solution? If yes, with which solution? Answer: The soil fertilizer was weighed and mixed with soil before filling in columns. Changed in the manuscript: We added a sentence to explain this. The soil was weighed, mixed perfectly with above fertilizer, respectively, and filled in its own column. Comment 12: - In figure 2: the authors draw 3 rock tablets, while the authors put only 2 rock tablets at the bottom of the column. Should change it. Changed in the manuscript: We have changed this. Comment 13: - Did the authors perform the same experiment without rock tablets if they leach their column in order to observe the leaching solution of the column? Answer: We didn't design that in this study. We didn't collect the soil solution. The leaching depended on the rainfall. Comment 14:- Did the authors put the 2 different rock tablets (calcite and dolomite) in the same column? Answer: Yes, we did. Comment 15: The authors should explain the reason of the fertilizer weight use in the experiment. Answer: Because the added amount of fertilizer can magnify and quicken the fertilization effect in the short-term according to another experiment from us, and can't affect the phenomenon we want to observe in this study. Another paper of ours (in preparation) about a series of different amount of fertilizer addition will discuss this issue. The added amounts of these 11 fertilizers were designed only by the average amount of N, P and K fertilizer in the local practical use. Changed in the manuscript: We have added the amount of N, P and K fertilizer in local practical use in this manuscript like this: N fertilizer: 160 kg NÂů ha-1; P fertilizer: 150 kg P2O5Âů ha-1; K fertilizer: 50 kg KÂů ha-1) Comment 16: 3. Results - L.164-165: Do not repeat Table 2 and Fig. 3. You may write: "The results are presented in Table 2 and in Figure 3. Changed in the manuscript: We have changed this. Comment 17: 4. Discussion - 4.1.: the first paragraph (L. 182-197) is quite general and it would be worthy to move it either in the introduction, or at least in the Materials and Methods section. Changed in the manuscript: We moved them to the introduction. Comment 18: 4.1. L.213-219: It is exactly the same text as in the introduction (L. 48-54) The authors may express their idea at least a little bit differently. Answer: Because they are for elaborating different problem, we think we should put one of them another way. Changed in the manuscript: We changed the statements in section 4.2. Comment 19: Information about soils and soil solutions are needed in order to understand their chemical evolution during the carbonate weathering. - Would it be possible to present the chemistry of each fertilizer used in this experiment? This can be added in supplementary information. Answer: yes, it is very important. Most of them have been discussed in section 4.2 and 4.3

Please also note the supplement to this comment:
http://www.earth-surf-dynam-discuss.net/esurf-2016-50/esurf-2016-50-AC3-supplement.pdf

**Supplement:**

[revised manuscript text omitted]

. We assume that the response of carbonate weathering to the addition of different fertilizer, such as N-fertilizer ($NH_4$ and $NO_3$),

P-fertilizer and Ca/Mg fertilizer, may display difference, which are so far poorly known, but likely significant. Here we sought to fully understand the agricultural impact on natural carbonate weathering, and to accurately evaluate the $CO_2$ consumption via carbonate weathering in agricultural areas.

Moreover, TtThe carbonate-rock-tablet test is used to determine the weathering rate of carbonate rock/mineral from the laboratory to the field (Chao et al., 2011; Chao et al., 2014; Dreybrodt et al., 1996; Gams, 1981; Gams, 1985; Jiang and Yuan, 1999; Liu and Dreybrodt, 1997; Plan, 2005; Trudgill, 1975)(Chao et al., 2014; Chao et al., 2011; Dreybrodt et al., 1996; Gams, 1981; Gams, 1985; Jiang and Yuan, 1999; Liu and Dreybrod, 1997; Plan, 2005; Trudgill, 1975)(Gams, 1981; Chao et al., 2011; Trudgill, 1975; Chao et al., 2014; Dreybrodt et al., 1996; Gams, 1985; Jiang and Yuan, 1999; Liu and Dreybrod, 1997; Plan, 2005). In the laboratory, the carbonate-rock-tablet is employed to study the kinetics of calcite dissolution/precipitation (Dreybrodt et al., 1996; Liu and Dreybrodt, 1997)(Dreybrodt et al., 1996; Liu and Dreybrod, 1997)(Dreybrodt et al., 1996; Liu and Dreybrod, 1997) and determine the rate of carbonate mineral weathering in the soil column (Chao et al., 2011)(Chao et al., 2011)(Chao et al., 2011). However, in the field, it is also used to observe the rate of carbonate weathering and estimated $CO_2$ consumption by carbonate weathering (Chao et al., 2014; Jiang and Yuan, 1999; Jiang et al., 2013; Plan, 2005)(Chao et al., 2014; Jiang and Yuan, 1999; Jiang et al., 2013; Plan, 2005)(Chao et al., 2014; Jiang and Yuan, 1999; Jiang et al., 2013; Plan, 2005). Although Liu (2011) argued that the carbonate-rock-tablet test may lead to the deviations of in estimated $CO_2$ consumption by carbonate weathering at the regional/global scale, in the cases ofwhere there are insufficient representative data (Liu, 2011)(Liu, 2011)(Liu, 2011), our results show thatyet iIt is nonetheless athe preferred optionmethod for the condition controlled contrastcomparative or stimulated experiment (Chao et al., 2011; Chao et al., 2014; Chao et al., 2017) (Chao et al., 2014; Chao et al., 2011)(Chao et al., 2011; Chao et al., 2014),. Where the result from the carbonate-rock-tablet test is consistent to the major element geochemical data of leachates from soil column(Chao et al., 2011).

ThusTherefore, in order to observe their difference between the impacts of different fertilizer addition on carbonate weathering in soil, a A field column experiment that involved embedding carbonate rock tablets with eleven different treatments was carried out in a typical karst area of southwest China, in order to observe the impacts of different fertilizer additions on carbonate weathering in soil.

**2. Materials and Methods**

**2.1 The study site**

This study was carried out in a typical karst area, namely the HuaXi Huaxi district District of Guiyang cityCity, Guizhou provinceProvince, SW China (26°23′N, 106°40′E, 1094 m aslASL). Guiyang, the capital city of Guizhou Province, is located in the central part of The the Provinceprovince, covering an area from 26°11′00″ to 26°54′20″N and 106°27′20″ to 107°03′00″E (aboutapproximately 8,000 km$^2$), with elevations ranging from 875 to 1655 m above mean sea levelASL. Guiyang has a population of more than 1.5 million people, a high wide diversity of karstic landforms, a high elevations and low latitude, with a subtropical warm-moist climate, and an average annual average temperature of 15.3 ℃ and annual precipitation of 1200 mm (Lang et al., 2006)(Lang et al., 2006)(Lang et al., 2006). A monsoonal climate often results in high precipitation during summer, with and much less during winter, although the humidity is often high during throughout most of the year (Han and Jin, 1996)(Han and Jin, 1996)(Han and Jin, 1996). Agriculture is a major land use in order to produce the vegetables and foods in the suburbs of Guiyang (Liu et al., 2006)(Liu et al., 2006)(Liu et al., 2006). The consumption of chemical fertilizer increased from

0.8 150 kg/ha Mt in 1980 to 1.0190 kg/ha Mt in 2013 (GBS, 2014)(GBS, 2014)(GBS,

2014).

**2.2 Soil properties**

The soil used in this column experiment was sampled from the B horizon (below cm in depth) of yellow-brown soil infrom  dug from a cabbage-corn or capsicum-corn rotation plantation in Huaxi dDistrict,. ItThe soil was air-dried, ground to pass through a 2-mm sieve, mixed thoroughly and used for the soil columns.

The soil pH ($V_{soil}$:$V_{water}$ = 1:2.5) were was determined by pH meter. The chemical characteristics of the soil, in-cluding organic matter (OM), $NH_4$-N, $NO_3$-N, available

P, available K, available Ca, available Mg, available S and available Fe, and available

S were determined according to the Agro Services International (ASI) Mmethod (Hunter, 1980)(Hunter, 1980)(Hunter, 1980),. OM was determined using an where the extracting solution used for O.M. containeding 0.2 mol $l^{-1}$ NaOH, 0.01 mol $l^{-1}$ EDTA,

% methanol, and 0.005 % Superfloc 127,. $NH_4$-N, $NO_3$-N, available Ca, and Mg were determined based using on an extraction extracting solution ofby  1 mol $l^{-1}$ KCl solution, whereas available K, P and Fe were determined using an extracted by extracting solution containing 0.25 mol $l^{-1}$ $NaHCO_3$, 0.01 mol $l^{-1}$ EDTA, 0.01 mol $l^{-1}$

$NH_4$F, and 0.005 % Superfloc 127,. Finally, and available S was determined using an extracting solution of ed by 0.1 mol $l^{-1}$ $Ca(H_2PO_4)_2$ and 0.005 % Superfloc 127. The results are shown in Table 1.

**2.3 Soil column and different fertilization treatments**

In order to test the hypothesis that the  impact of different chemical fertilizers on carbonate weathering may be different, columns ($\emptyset$ = 20 cm,

H= 15 cm) were constructed from 20 cm diameter polyvinylchloride (PVC) pipe (Fig.

2). A hole ($\emptyset$ = 2 cm)  was  placed at the bottom of each column to discharge soil water from the  soil column. A polyethylene net  ($\emptyset$ 0.5 mm)

was placed in the bottom of the columns to prevent  soil loss .

A 2 cm thick filter  layer,  including gravel, coarse sand and fine sand, was spread over the net.  Two different carbonate rock tablets were buried in the bottom of each soil column (Fig. 2). Based on the common kinds of chemical fertilizers and the main objective of this study, eleven fertilization treatments, each with three replicates, were set up in the field column experiment.

There are : (1) control without fertilizer (CK); (2) 43g $NH_4NO_3$ fertilizer (CF); (3) 85g $NH_4HCO_3$ fertilizer (NHC); (4) 91g $NaNO_3$ fertilizer (NN); (5) 57g

$NH_4Cl$ fertilizer (NCL); (6) 51g $(NH_4)_2CO_3$ fertilizer (NC); (7) 52g $Ca_3(PO_4)_2$

fertilizer (CP); (8) 15g $(NH_4)_3PO_4$ fertilizer (NP); (9) 44g fused calcium-magnesium phosphate fertilizer (Ca-Mg-P); (10) 32g Urea fertilizer (U); and (11) 10g $K_2CO_3$

fertilizer (PP).

An aliquot of 6 kg of soil was weighed (bulk density = 1.3 g/cm$^3$), mixed throughly with one of the above fertilizers, and filled into its own column. This process was repeated for all three replicates of the 11 fertilizer treatments. These soil columns were placed at the field experiment site in Huaxi District, Guiyang of Southwestern China for a whole year.

**2.4 The rate of carbonate weathering**

Two different kinds of carbonate rock tablets (2 cm × 1 cm × 0.5 cm in size) were established placed in the bottom of each soil column to explore examine the rate of carbonate weathering in the soil. The two different kinds of carbonate rock collected from the karst area of Huaxi dDistrict were: (1) limestone with 60-65 % micrite,

30-35 % microcrystalline calcite, and 2-3 % pyrite; and (2) dolostone with 98-99 %

power fine crystalline dolomite, 3-5% microcrystalline calcite, 1 % pyrite, and little trace quantities organic matter. All of the tablets were baked heated at 80 ℃ for 4

hours, then weighed in a 1/10000 electronic balance in the laboratory, tied to a labeled by tying a label with fishing line, and then buried at the bottom of each soil column.

After a whole year, Tthey tablets were taken outremoved carefully, rinsed, baked and weighed after a whole year.

The amount of weathering carbonate weathering (A$_{ew}$), the ratio of carbonate weathering (R$_{ew}$) and the rate of carbonate weathering (R$_{aew}$) for limestone and dolomite were calculated according to the weight difference of the tablets using the following formulas:

$$A_{ew} = (W_i - W_f) \qquad\qquad (1)$$

$$R_{ew} = (W_i - W_f)/W_i \qquad\qquad (2)$$

$$R_{aew} = (W_i - W_f)/(S*T) \qquad\qquad (3)$$

where W$_i$ is the initial weight of the carbonate rock tablets, W$_f$ is their final weights,

S is the surface area of carbonate  rock tablet, and T is the length of the experiment period.

**2.5 Statistical analysis**

Statistical analysis was performed using IBM SPSS 20.0 (Statistical Graphics Crp, Princeton, USA). All results of carbonate weathering were reported as the means ± standard deviations (SD) for the three replication.

**3. Results**

3.1 Weathering rate of carbonate under different fertilized treatments

The  and  of limestone and dolostonewein Table 2.  The results showed that the A$_{cw}$, R$_{cw}$  of limestone  under urea, NH$_4$NO$_3$, NH$_4$Cl, (NH$_4$)$_2$CO$_3$ and NH$_4$HCO$_3$  treatments were 8.48 ±0.96, 6.42 ±0.28, 5.54 ±0.64, 4.44 ±0.81 and 4.48 ±0.95 ‰  significantly greater than that under the control treatment 0.48 ±0.14 ‰ (see Fig. 3). In addition, the observed R$_w$ of  dolostone were 6.59 ±0.67, 5.30 ±0.87, 4.77 ±0.78, 4.94 ±1.91 and 3.22 ±0.87 ‰ respectively, under these same five fertilization treatments, in contrast to

-0.31 ± 0.09 ‰ in the control treatment). This clearly demonstrates that the addition of these five fertilizers increased the rate of carbonate weathering.

The remaining treatments made no significant differences in the $R_w$ and $R_{aw}$ of limestone and dolostone in comparison  to the control treatment (Fig. 3).  In the $(NH_4)_3PO_4$ treatment, the $R_w$ were only 1.08 ± 0.34 ‰,  and 0.75 ± 0.21‰ for limestone and dolomite, respectively, while the $R_{aw}$ were 4.00 ± 1.15 g m$^{-2}$ a$^{-1}$ and 1.00 ± 1.01  g m$^{-2}$ a$^{-1}$ for limestone and dolomite, respectively. These values are less than those under the other four $NH_4$-fertilizers as mentioned above. The  $R_w$ and $R_{aw}$ in the $NaNO_3$ treatment failed to show notable difference with the control treatment, exhibiting little effect of the $NaNO_3$ fertilizer addition on carbonate weathering (Fig. 3).

Except for the $R_w$ of limestone approaching zero in the $Ca_3(PO_4)_2$ treatment , all the values of  $R_w$ and $R_{aw}$  in Ca-Mg-P  $K_2CO_3$ and $Ca_3(PO_4)_2$ treatments showed  negative value. This indicates that the addition of Ca-Mg-P $K_2CO_3$ and $Ca_3(PO_4)_2$ fertilizers led to precipitation at the surface of the carbonate mineral, which can be explained by common ion effect.

3.2 Comparison of limestone of dolomite

In Fig. 4, we plotted the $R_w$ of limestone vs.

dolomitestone tablets in a, we plotted Fig. 4, a linear correelation diagram, in order to compare the weathering responses of with the R*w* of limestone vs. dolostone tabletslimestone with dolostone. The results shows that the R*w* of limestone and dolostone exhibits a high positive correlaetion ($R^2$=0.9773,; see Fig. 4), suggestindicating that the weathering of ANOVA was use the limestone and dolostone are similar under different treatments were similar. Thus, we will explain the results with in terms of carbonates, rather than instead of 
[revised manuscript text omitted]

Calmels, D., J. Gaillardet, and L. François. 2014. Sensitivity of carbonate weathering to soil $CO_2$ production by biological activity along a temperate climate transect. Chem Geol 390:74-86.

Chao, S., L. Changli, Z. Yun, and H. Hongbing. 2014. Impact of animal manure addition on agricultural lime weathering in acidic soil: pH dependence and $CO_2$ independence of agricultural lime weathering. Procedia Earth and Planetary Science:405-409.

Chao, S., L. Changli, W. Junkun, Z. Yun, and H. Hongbing. 2011. Impact of the Addition of a

Compound Fertilizer on the Dissolution of Carbonate Rock Tablets: a Column Experiment. Appl Geochem 26:170-173.

Chien, S.H., L.I. Prochnow, S. Tu, and C.S. Snyder. 2011. Agronomic and environmental aspects of phosphate fertilizers varying in source and solubility: an update review. Nutr Cycl Agroecosys 89:229-255.

Chou, L., R.M. Garrels, and R. Wollast. 1989. Comparative study of the kinetics and mechanisms of dissolution of carbonate minerals. Chem Geol 78:269-282.

Dreybrodt, W. 1988. Processes in karst systems: physics, chemistry, and geology Springer-Verlag Berlin Heidelberg.

Dreybrodt, W., J. Lauckner, L. Zaihua, U. Svensson, and D. Buhmann. 1996. The kinetics of the reaction $CO_2 + H_2O \rightarrow H^+ + HCO_3^-$ as one of the rate limiting steps for the dissolution of calcite in the system $H_2O$-$CO_2$-$CaCO_3$. Geochim Cosmochim Ac 60:3375-3381.

Egli, M., and P. Fitze. 2001. Quantitative aspects of carbonate leaching of soils with differing ages and climates. Catena 46:35-62.

Errin, A.S.P., A.P. Robst, and J.L.P. Robst. 2006. Impact of nitrogen fertilizers on natural weathering processes : Evident role on $CO_2$ consumption The oxygen isotopic composition of Precambrian cherts, pp. 2332-2332 Goldschmidt Conference, Vol. 58.

Etchanchu, D., and J. Probst. 1988. Evolution of the chemical composition of the Garonne River water during the period 1971-1984. Hydrological sciences journal 33:243-256.

(ed.) 1981. Proceedings of the 8th International Congress of Speleology Bowling Green, Kentucky.

Gams, I. 1985. International comparative measurements of surface solution by means of standard limestone tablets. Razpr IV Razeda Sazu 26:361-386.

Gandois, L., A.S. Perrin, and A. Probst. 2011. Impact of nitrogenous fertiliser-induced proton release on cultivated soils with contrasting carbonate contents: A column experiment. Geochim Cosmochim Ac 75:1185-1198.

GBS. 2014. 2014 Guizhou Statistical Yearbook Statistical Press of China, Beijing.

Hagedorn, B., and I. Cartwright. 2009. Climatic and lithologic controls on the temporal and spatial variability of $CO_2$ consumption via chemical weathering: An example from the Australian Victorian Alps. Chem Geol 260:234-253.

Hamilton, S.K., A.L. Kurzman, C. Arango, L. Jin, and G.P. Robertson. 2007. Evidence for carbon sequestration by agricultural liming. Global Biogeochem Cy 21:1-12.

Han, Z., and Z. Jin. 1996. Hydrogeology of Guizhou Province, China Seismic Publication, Beijing.

Hunter, A.H. 1980. Laboratory and greenhouse techniques for nutrient survey to determine the soil amendments required for optimum plant growth. Mimeograph. Agro Service International (ASI), Florida, USA.

Jiang, Y. 2013. The contribution of human activities to dissolved inorganic carbon fluxes in a karst underground river system: Evidence from major elements and $\delta^{13}C_{DIC}$ in Nandong, Southwest China. J Contam Hydrol 152:1-11.

Jiang, Y., Y. Wu, C. Groves, D. Yuan, and P. Kambesis. 2009. Natural and anthropogenic factors affecting the groundwater quality in the Nandong karst underground river system in Yunan, China. J Contam Hydrol 109:49-61.

Jiang, Z., and D. Yuan. 1999. $CO_2$ source-sink in karst processes in karst areas of China. Episodes 22:33-35.

Jiang, Z., Y. Lian, and X. Qin. 2013. Carbon cycle in the epikarst systems and its ecological effects in

South China. Environ Earth Sci 68:151-158.

Kiefer, R.H. 1994. Temporal cycles of karst denudation in northwest Georgia, USA. Earth Surf Proc

Land 19:213-232.

Kitano, Y., M. Okumura, and M. Idogaki. 1978. Uptake of phosphate ions by calcium carbonate.

Geochem J:29-37.

Kump, L.R., S.L. Brantley, and M.A. Arthur. 2000. Chemical weathering, atmospheric $CO_2$, and climate. Annu Rev Earth Pl Sc 28:611-667.

Lang, Y., C. Liu, Z. Zhao, S. Li, and G. Han. 2006. Geochemistry of surface and ground water in

Guiyang, China: Water/rock interaction and pollution in a karst hydrological system. Appl Geochem

21:887-903.

Lerman, A., and L. Wu. 2006. $CO_2$ and sulfuric acid controls of weathering and river water composition. J Geochem Explor 88:427-430.

Lerman, A., L. Wu, and F.T. Mackenzie. 2007. $CO_2$ and $H_2SO_4$ consumption in weathering and material transport to the ocean, and their role in the global carbon balance. Mar Chem 106:326-350.

Li, S., D. Calmels, G. Han, J. Gaillardet, and C. Liu. 2008. Sulfuric acid as an agent of carbonate weathering constrained by $\delta^{13}C_{DIC}$: Examples from Southwest China. Earth Planet Sc Lett

270:189-199.

Li, S., Z. Xu, H. Wang, J. Wang, and Q. Zhang. 2009. Geochemistry of the upper Han River basin,

China: 3: Anthropogenic inputs and chemical weathering to the dissolved load. Chem Geol 264:89-95.

Lian, B., Y. Chen, L. Zhu, and R. Yang. 2008. Effect of Microbial Weathering on Carbonate Rocks.

Earth Science Frontiers 15:90-99.

Liu, C., S. Li, Y. Lang, and H. Xiao. 2006. Using $\delta^{15}N$ and $\delta^{18}O$ Values To Identify Nitrate

Sources in Karst Ground Water, Guiyang, Southwest China. Environ Sci Technol 40:6928-6933.

Liu, Z. 2011. " Method of maximum potential dissolution" to calculate the intensity of karst process and the relevant carbon sink: With discussions on methods of solute load and carbonate-rock-tablet test.

Carsologica Sinica 30:79-82.

Liu, Z., and W. Dreybrod. 1997. Dissolution kinetics of calcium carbonate minerals in $H_2O$-$CO_2$

solutions in turbulent flow: The role of the diffusion boundary layer and the slow reaction $H_2O + CO_2$

$\leftrightarrow H^+ + HCO_3^-$. Geochim Cosmochim Ac 61:2879-2889.

Liu, Z., W. Dreybrodt, and H. Liu. 2011. Atmospheric $CO_2$ sink: silicate weathering or carbonate weathering? Appl Geochem 26:292-294.

Liu, Z., H. Sun, L. Baoying, L. Xiangling, Y. Wenbing, and Z. Cheng. 2010. Wet-dry seasonal variations of hydrochemistry and carbonate precipitation rates in a travertine-depositing canal at

Baishuitai, Yunnan, SW China: Implications for the formation of biannual laminae in travertine and for climatic reconstruction. Chem Geol 273:258-266.

NBS. 2014. 2014 Statistical Yearbook. Statistical Press of China, Beijing.

Oh, N., and P.A. Raymond. 2006. Contribution of agricultural liming to riverine bicarbonate export and

$CO_2$ sequestration in the Ohio River basin. Global Biogeochem Cy 20:1-17.

Perrin, A., A. Probst, and J. Probst. 2008. Impact of nitrogenous fertilizers on carbonate dissolution in small agricultural catchments: Implications for weathering $CO_2$ uptake at regional and global scales.

Geochim Cosmochim Ac 72:3105-3123.

[revised manuscript text omitted]

Acw - the amount of carbonate weathering; Rew - the ratio of carbonate weathering; Raew - the rate of
carbonate weathering; Rw =1000 (Wi-Wf)/Wi, and Rew = (Wi-Wf)/Wi, Raew = (Wi-Wf)/(S*T),
where Wi is the initial weight of the carbonate rock tablets, and Wf is the final weight. S is the surface
area of carbonate rock tablets here, we used S = 7 cm² for every
tablets), and T is the experiment period. Values are reported as means ± standard deviations, n = 3.
Values in each column followed by different letters are significantly (p <0.05) different based on
one-way ANOVA.

~~Table 3: The main reaction and effects in fertilized treatments, and the potential nitrogenous~~
~~transformation (The amount of generated $NH_4^+$ $NO_2^-$) at the initial phase of the experiment~~

Table 3. The main reaction and effects in these 11 fertilizer treatments

| Treatment | Main reactions and effects |
|---|---|
| 1. Control | $Ca_{(1-x)}Mg_xCO_3 + CO_2 + H_2O \rightarrow (1-x)\ Ca^{2+} + xMg^{2+} + 2HCO_3^-$ |
| 2. NH4NO3 | $2Ca_{(1-x)}Mg_xCO_3 + NH_4NO_3 + 2O_2 \rightarrow 2(1-x)\ Ca^{2+} + 2xMg^{2+} + 2NO_3^- + H_2O + 2HCO_3^-$ |
| 3. NH4HCO3 | $NH_4HCO_3 \rightarrow NH_3\uparrow + H_2O + CO_2\uparrow$
 $2Ca_{(1-x)}Mg_xCO_3 + NH_4HCO_3 + 2O_2 \rightarrow 2(1-x)\ Ca^{2+} + 2xMg^{2+} + NO_3^- + H_2O + 3HCO_3^-$ |
| 4. NaNO3 | $Ca_{(1-x)}Mg_xCO_3 + NaNO_3 + CO_2 + H_2O \rightarrow (1-x)\ Ca^{2+} + xMg^{2+} + Na^+ + NO_3^- + 2HCO_3^-$ |
| 5. NH4Cl | $2Ca_{(1-x)}Mg_xCO_3 + NH_4Cl + 2O_2 \rightarrow 2(1-x)\ Ca^{2+} + 2xMg^{2+} + NO_3^- + Cl^- + H_2O + 2HCO_3^-$ |
| 6. (NH4)2CO3 | $(NH_4)_2CO_3 \rightarrow 2NH_3\uparrow + H_2O + CO_2\uparrow$
 $3Ca_{(1-x)}Mg_xCO_3 + (NH_4)_2CO_3 + 4O_2 \rightarrow 3(1-x)\ Ca^{2+} + 3xMg^{2+} + 2NO_3^- + 2H_2O + 4HCO_3^-$ |
| 7. Ca3(PO4)2 | (1) $(1-x)\ Ca^{2+} + xMg^{2+} + 2HCO_3^- \rightarrow Ca_{(1-x)}Mg_xCO_3 + CO_2 + H_2O$
 (2) $Ca + PO_4 \rightarrow Ca-P$ |
| 8. (NH4)3PO4 | (1) $2Ca_{(1-x)}Mg_xCO_3 + NH_4^+ + 2O_2 \rightarrow 2(1-x)\ Ca^{2+} + 2xMg^{2+} + NO_3^- + H_2O + 2HCO_3^-$
 (2) $Ca + PO_4 \rightarrow Ca-P$ |
| 9. Ca-Mg-P | (1) $(1-x)\ Ca^{2+} + xMg^{2+} + 2HCO_3^- \rightarrow Ca_{(1-x)}Mg_xCO_3 + CO_2 + H_2O$
 (2) $Ca + PO_4 \rightarrow Ca-P$ |
| 10. Urea | $3Ca_{(1-x)}Mg_xCO_3 + CO(NH_2)_2 + 4O_2 \rightarrow 3(1-x)\ Ca^{2+} + 3xMg^{2+} + 2NO_3^- + 4HCO_3^-$ |
| 11. K2CO3 | (i) $(1-x)\ Ca^{2+} + xMg^{2+} + 2HCO_3^- \rightarrow Ca_{(1-x)}Mg_xCO_3 + CO_2 + H_2O$ |

$$(ii)\ K_2CO_3 + H_2O \rightarrow 2K^+ + HCO_3^- + OH^-$$

Note: (1) Common ion effect: The $Ca_{(1-x)}Mg_xCO_3$ produces when the concentrations of $Ca^{2+}$, $Mg^{2+}$ and/or $HCO_3^-$ increases (for treatment 7, 9 and 11): $(1-x)\ Ca^{2+} + xMg^{2+} + 2HCO_3^- \rightarrow Ca_{(1-x)}Mg_xCO_3 + CO_2 + H_2O$; (2) Inhibition of  calcite dissolution/precipitation by phosphate: calcium orthophosphate (Ca-P) precipitation produces on the surface of calcite after the addition of $PO_4^{3-}$ in soil, resulting in  inhibition of the dissolution/precipitation of calcite (for treatment 7, 8 and 9): $Ca + PO_4 \rightarrow Ca\text{-}P$

Table 4: The amount of fertilizer-derived $NH_4^+$ at the initial phase of the experiment and the potential nitrogenous transformation ($NH_4^+$-$NO_3^-$)

| Treatment | Molecular mass g/mol | Amount of added fertilizer /g | Molar amount /mole | Amount of fertilizer-derived $NH_4^+$ /mole | The maximum of N products /mole |
|---|---|---|---|---|---|
| $NH_4NO_3$ | 80 | 43 | 0.54 | 0.54 | 1.08 |
| $NH_4HCO_3$ | 79 | 85 | 1.08 | 1.08 | 1.08 |
| $NaNO_3$ | 85 | 91 | 1.07 | 0.00 | 1.07 |
| $NH_4Cl$ | 53.5 | 57 | 1.07 | 1.07 | 1.07 |
| $(NH_4)_2CO_3$ | 96 | 51 | 0.53 | 1.06 | 1.06 |
| $Ca_3(PO_4)_2$ | 310 | 52 | 0.17 | 0.00 | 0.00 |
| $(NH_4)_3PO_4$ | 149 | 15 | 0.10 | 0.30 | 0.30 |
| Ca-Mg-P | nd | 44 | nd | 0.00 | 0.00 |
| Urea | 60 | 32 | 0.53 | 1.06 | 1.06 |
| $K_2CO_3$ | 138 | 10 | 0.07 | 0.00 | 0.00 |

nd= no data. The amount of added fertilizer (g) divided by its molecular mass (g/mol) is the molar amount of fertilizer (mole). The amounts of fertilizer-derived $NH_4^+$ is calculated by their own ionization or hydrolysis processes. The maximum of N products is estimated by their main reactions in Table 3.

[Figure]

Fig. 1 The change  in chemical fertilizer consumption in China during the 1980-2013 period
The data were collected from National Bureau of Statistics of the People's Republic of China
(NBS, 2014) (http://www.stats.gov.cn/tjsj/ndsj/)

[Figure]

Fig. 2 Sketch  of the soil column with rock tablets

[Figure]

[Figure]

[Figure]

Fig. 3 The R*w* (‰) of limestone and dolostone under different  fertilizer treatment

 Treatment 1- Control; 2- $NH_4NO_3$; 3- $NH_4HCO_3$; 4- $NaNO_3$; 5- $NH_4Cl$; 6-  $(NH_4)_2CO_3$; 7- $Ca_3(PO_4)_2$; 8-  $(NH_4)_3PO_4$; 9- Ca-Mg-P; 10- Urea; 11- $K_2CO_3$. R*w* = 1000(W*i*-W*f*)/W*i*, where W*i* is the initial weight of the carbonate rock tablet, and W*f* is the final weight.

[Figure]

[Figure]

Fig. 4 The linear correlation of R$w$ (‰) of limestone and dolostone

R$w$ = 1000(W$i$-W$f$)/W$i$, where W$i$ is the initial weight of the limestone tablets, and W$f$ is their

                          final weight.

[Figure]

Fig. 4 5 The R*w* (‰) of limestone ratio of limestone weathering and the molar amount of produced $NH_4^+$ under different fertilization fertilizer treatments

Treatment 1- Control; 2- $NH_4NO_3$; 3- $NH_4HCO_3$; 4- $NaNO_3$; 5- $NH_4Cl$; 6 ( $(NH_4)_2CO_3$; 7- $Ca_3(PO_4)_2$; 8 ( $(NH_4)_3PO_4$; 9- Ca-Mg-P; 10- Urea; 11- $K_2CO_3$. R*e*w = 1000(W*i*-W*f*)/W*i*, where W*i* is the initial weight of the lim*s*estone tablet*s*, and W*f* is the*ir* final weight.

**校对报告**

当前使用的样式是 【Chemical Geology】
当前文档包含的题录共99条

有0条题录存在必填字段内容缺失的问题
所有题录的数据正常

---

## Referee Report (RR1)

**Review of the paper esurf-2016-50.**

This paper is dealing with the role of the application of chemical and organic fertilizers on carbonate weathering in a karst area. Their approach was based on a laboratory experiment using a soil column including two carbonate (limestone and dolostone) rock tablets over one year on the field. The authors discussed the loss (or gain) of weight of each rock tablets in term of variability of carbonate weathering under various fertilizer treatments. The authors concluded that the main NH4 fertilizers induced the most important carbonate weathering, while the other fertilizers have a much lesser effect. The topic as such should be well suited for a publication in Earth Surface Dynamics, but this manuscript is not, at least in its present state. Compared to the first version, there are some improvements, but some parts need some clarifications. I would suggest minor revisions of the present manuscript.

Here are my comments and suggestions:

In some paragraphs, the English writing may be slightly improved.

In the field, how are placed the different experimental columns? Did you have a random position, mixing the different modalities?

The authors mixed "perfectly" the chemical fertilizers with the sieved soils. It was not directly spread on the field/columns. This may be an artefact compared to the natural field, where fertilizers are spread on the soils, relatively far from the rock. In the last part of the discussion, this point may be highlighted, as the authors compared their data with the literature… This way, their weathering rate may be (slightly) overestimated. This other point is that in the literature most studies approach the weathering estimate from riverine data as the authors discussed.

- L41 – 42 : "(…) processes including the reaction between carbonates and  protons derived (…)".
- L 45 and 47: you may be more specific on the origin of sulfuric acid, and the role of acidic soil in the carbonate – proton relationship. "Acidic soil" is still too broad.
- L55, you may add the increase proportion of mineral fertilizers (increase by 365%), in order to compare with the 3.3% worldwide increase… I was wondering what was the cause of such sharp increase in chemical fertilizers consumption. Is it just an effect of the increase of the size of agricultural land or is it a consequence of a change of fertilizer habit (more NO3 or NH4 fertlizers)?
- L118-119: You worked on Guizhou area where the consumption of chemical fertilizers increased by about 26% (far less than the increase of fertilizer consumption at the scale of the whole country).
- L121: Could you be more specific for the soil classification, more precise than B horizon?
- L123: did you crush (ground) the sample before to pass it through 2 mm sieve? Or did you pass the air-dried soil through 2 mm sieve, and after you crushed it… this may seem to be a detail, but it is really important.
- L141: Did you have any silt or clay loss with a 0.5 mm net at the bottom of your column? Did you add some quartz wool in addition to the sand "filter" above the PE net?
- L142-143: "Two different carbonate rock tablets were buried in the bottom of each soil column"; by this, do you mean that you put only one tablet of two different carbonate rocks? So the carbonate rocks are different, or are they two aliquots of the same rock type?
- L145-150: How did you determine the weight of each chemical fertilizer applied on each soil experiment?

- L151-152: you did mix each fertilizer with air-dried soils. So the fertilizer was not spread on the column? As soil and fertilizers were perfectly mixed, did you also crush the chemical fertilizers or did you leave them in their original shapes?
- L158-159: you used 2 carbonate types: a limestone and a dolostone. Did you put one tablet of each carbonate in the same column, or did you put 2 tablets of the same carbonate (in order to have duplicates) in the same column?
- L176: This is at this point that we know that you have triplicates. How did you obtain these triplicates: did you use different columns for replicates, or are they in the same columns… It is important in order to understand on which data you performed a statistical analysis. How many columns did you in total?
- You should add "respectively" at the end of your sentence when you're listing some results…
- L189: the "rest treatments"? do you mean the other chemical fertilizers other than $NH_4NO_3$, $NH_4Cl$, $(NH_4)_2CO_3$, $NH_4HCO_3$?
- L206: you explain that there is no difference in the weathering behavior of limestone and dolostone, however the control experiment show different behavior. Did you test this similarity?
- L213-215: you may present these 3 equations in the following order: (5) > (4) > (3), following an increasing pKa (or pH).
- L218: $H_2CO_3$ is not only formed in the soil, it is first formed in the atmosphere with the dissolution of atmospheric $CO_2(g)$ into rain droplets and rain, as $CO_{2aq}$ and then $H_2CO_3$. The concentration of $H_2CO_3$ is exacerbated into soil because of the presence of organic $CO_2$ from respiration. And yes, one of the main control of carbonate dissolution is the amount of rainfall. That's why is it important to know how the different columns were placed in the field, randomly or the same modalities were placed together (L223-224). If randomly, you can say that rainfall may not be consider an a controlling effect for your experiment.
- L231-232: you may be more precise for "CO2 degassing"… Issued from carbonate dissolution? For respiration?
- L244: the year of the publication from Singh et al is missing.
- L305: you did calculate the initial fertilizer –derived NH4 per unit. But it would be interesting to have the initial rate of fertilizer spread in the field and to compare it with what it is really applied in the Chinese agricultural watersheds.
- L396-397:
- Table 2: what is the significance of a, b, c behind the numbers (ANOVA)?

---

## Author Response (ED1)

**Responses to Referee 1**

**Comment 1:** Could authors provide the ratio of the nitrate fertilizer vs. total nitrogen fertilizers in studied area, or China, or whole world? It is important for the significance of the manuscript and the experiment.

Answer: Yes, it is very important, but we can't find the specific published data about it. Only one figure we can calculate according to a reference published in 2008 is that the global production of $NH_4NO_3$ accounts for 10% of the total N fertilizers, and about 4% in China. But we think this figure is not fit to be cited in this manuscript. We think that this study only focused on the relative potential mechanism, and the study on the estimation of the impacts will be considered in the future.

**Comment 2:** Could authors compare results with data by Prof. Yuan DX's group paper ?

Answer: We have cited the relative result from Prof. Yuan Dx's group in this study, such as the papers from Jiang Z., et al.; Liu Z., et al. and Jiang Y., et al.. In fact, we are familiar with them and their study focuses, we know that their studies are a little bit different from ours so far.

**Comment 3:** The manuscript need more detail for the experiment. L134-L139: Please give a detail introduction for the added amount of fertilizers in these treatments. It seems that the added amount of nitrogen is slight difference.

(1) What's the proportion of these eleven fertilization treatments in local practical use?

Answer: The added amounts of these 11 fertilizers were designed only by the average amount of N, P and K fertilizer in the local practical use.

Changed in the manuscript: We have added the amount of N, P and K fertilizer in local practical use in this manuscript like this: N fertilizer: 160 kg N $\cdot$ha$^{-1}$; P fertilizer: 150 kg $P_2O_5$ $\cdot$ha$^{-1}$; K fertilizer: 50 kg K $\cdot$ha$^{-1}$)

(2) Why choose the added amount of fertilizers are 30 times than its local practical amount? The application fertilizers in local practical may change two or three times to use. Do you think the added fertilizers by one time may affect the result?

Answer: Because the added amount of fertilizer can magnify and quicken the fertilization effect in the short-term according to another experiment from us, and can't affect the phenomenon we want to observe in this study. Another paper of ours (in preparation) about a series of different amount of fertilizer addition will discuss this issue.

(3)Why don't authors set different height for this experiment, which might be more interesting?

Answer: Thank you for your good suggestion. This study is conducted as a simply start. The suggestion on different height must be interesting. We are considering that in further study.

(4)Does author consider that the land use influences the carbonate weathering in the experiment?

Answer: In fact, there are some good studies to be published about that around the world. But most of them are conducted by the search and evaluation of riverine hydro-geochemical data. Because of this, we did this study from another angle, and hope to connect them in the future.

**Comment 4:** The authors made three replications. So please show the data errors for each average value.

Answer: Fig.3 has showed these error bars. We will add them in Table 2.

Changed in the manuscript: The error data have been added in Table 2.

**Comment 5:** Could you assess the variation of nitrate fertilizer change in the column. Then understand the balance between acid producing and carbonate weathering together.

Answer: We tried our best to explain the chemical process of N fertilizer in section 4.3 with relative chemical reactions.

**Comment 6:** Line 15-17, the sentence has too many "different"s. Please revise it.

Changed in the manuscript: We use "these discrepancies" instead of "their differences".

**Comment 7:** In section 2.2, could authors provide details about abbreviation of OM, ASI method and others when you write that at the first time. Please check that in whole manuscript.

Changed in the manuscript: We have changed this.

**Comment 8:** L162-166 It seems Table 2 and Fig3 are repeated.

Changed in the manuscript: We deleted the relative data in Table 2, and leave the Fig. 3 because the figure is easier to see. And we also changed corresponding texts.

**Comment 9:** L182-L197 This paragraph can be removed to the introduction.

Changed in the manuscript: We moved them to the introduction.

**Comment 10:** L213-L219 It is repeated the introduction (L49-L54).

Answer: Because they are for elaborating different problem, we think we should put one of them another way.

Changed in the manuscript: We changed the statements in section 4.2.

**Comment 11:** Major conclusion might be revised. Ammonium fertilizer mainly includes $NH_4NO_3$, $NH_4Cl$, $(NH_4)_2CO_3$ fertilizers, not includes urea fertilizer. I suggest reductive nitrogenous fertilizer could enhance carbonate weathering via nitrification.

Changed in the manuscript: Yes, the statement that nitrogenous fertilizer can aid carbonate weathering should be replaced by ammonium fertilizer" in this manuscript is not precise. We deleted it.

**Responses to Referee 2**

**Comment 1** - The authors did not present very well the process/method of weathering which has been used in this experiment:   (1)did the authors perform a leaching of the soil column? How are the fertilizers introduced in the soil column? Are spread mixed with soil or spread in solutions? The lack of explanation of the method used does not allow us to assess the results at their fair value. There is also a lack of discussion and comparison of numerical values obtained in other experiments and in natural and agricultural catchments. The carbonate weathering is only estimated based on the weight of each rock tablets. It is not checked by the geochemistry of both rock tablets and the potential weathering/soil solution. Indeed, it would have been interesting to have an estimation of the chemical weathering.

Answer: The fertilizer was mixed with soil before filling in columns.

Changed in the manuscript: We added a sentence to explain this. The soil was weighed, mixed perfectly with above fertilizer, respectively, and filled in its own column.

**Comment 2** – To speed up the carbonate weathering, the fertilizers were introduced by increasing their amount by 30 times (Why 30 times?). It is a bit problematic, because the authors changed the soil/fertilizers ratio compared to "natural/anthropogenic" ratio? What is this ratio in the local agricultural catchments? What are the specificities these local catchments compared to national Chinese catchments and worldwide catchments?

Answer: Because the added amount of fertilizer can magnify and quicken the fertilization effect in the short-term according to another experiment from us, and can't affect the phenomenon we want to observe in this study.   Another paper of ours (in preparation) about a series of different amount of fertilizer addition will discuss this issue. The added amounts of these 11 fertilizers were designed only by the average amount of N, P and K fertilizer in the local practical use.

Changed in the manuscript: We have added the amount of N, P and K fertilizer in local practical use in this manuscript like this: (N fertilizer: 160 kg N $\cdot ha^{-1}$; P fertilizer: 150 kg $P_2O_5$ $\cdot ha^{-1}$; K fertilizer: 50 kg K $\cdot ha^{-1}$)

**Comment 3** – The variability of the experimental replicates should be shown (average and standard deviations), presented and discussed. This can be presented in Table 2.

Answer: We did it.

**Comment 4** – In general, the authors used limestone and dolostone tablets. They did not discuss the results of dolostone tablets, only those from limestone tablets. In the discussion, the difference or similarity between dolostone and limestone is erased as the authors discuss about carbonates. More attention, or at least an explanation about the use of the general term of "carbonates" instead of the difference between dolostone and limestone should be given.

Answer:   The difference between limestone and dolostone is not noteworthy, so we use carbonate instead. Yes, we need to give some sentences to explain this.

Changed in the manuscript: We added the statement "The result between limestone and dolostone weathering under different fertilization treatment were similar. We will explain the results with carbonates instead of individual dolostone and limestone." in this manuscript to explain

**Comment 5:** In several times in the manuscript (last sentence of the abstract, first paragraph of the results, and the last sentence of the conclusion) the authors used the expression "can aid carbonate weathering": they should precise if the fertilizers enhance, increase, or decrease carbonate weathering.

Changed in the manuscript: The statement that nitrogenous fertilizer can aid carbonate weathering should be replaced by ammonium fertilizer" in this manuscript is not precise. We deleted it.   And we replaced the rest aids with the word "increase".

**Comment 6:** Introduction: - L.43 - The authors should add references showing the relationship between carbonate weathering and climate in addition to Liu et al. (2010, 2011); for example Kump et al., 2000). –

Changed in the manuscript: We added it.

**Comment 7:** L.47 - The authors should precise that the disturbance of $CO_2$ consumption disturbance may be overestimated at a local scale by taking into account $Ca^{2+}$ and $Mg^{2+}$ produced by a natural carbonate weathering and those produced indirectly by anthropogenic activities in the watershed. And what about this disturbance at a global scale?

Answer: Here, we are just trying to introduce the potential disturbance at the regional/global scales by summarizing and classifying some references in the 1st paragraph. And the specific disturbances from fertilizer addition were further discussed in the 2nd paragraph.

**Comment 8:** 2.2. Soil properties : - At which depth did the authors sample their soils? - Should precise pH(H2O) - Precise what OM means: organic matter I suppose. - Precise what ASI method means. - What is the soil typology?

Answer: The pH had been listed in Table 1.

Changed in the manuscript: The meanings of OM and ASI have been added. We changed the statement "The soil used in this column experiment was sampled from the B horizon (below 20 cm in depth) of yellow-brown soil in a cabbage-corn or capsicum-corn rotation plantation in Huaxi district." to explain the soil samples and typology.

**Comment 9:** 2.3. Soil column - What is the filter material?

Answer: Yes, it is a misleading expression here.

Changed in the manuscript: It has been changed into: A Polyethylene net (Ø 0.5 mm) was placed in the bottom of the columns to prevent soil loss. A filter sand layer with 2 cm thickness including gravel, coarse sand and fine sand was spread on the net.

**Comment 10:** What kind of carbonate rocks did the authors use for their experiment? Are they reference rocks or rocks from karst area of HuaXi district?

Answer: yes, it was collected from karst area of Huaxi district.

Changed in the manuscript: We added this information in this manuscript.

**Comment 11:** How did the authors deposit each fertilizer in the column? In liquid or solid form? At which temperature has the experiment been performed? - Did you leach the soil column with a solution? If yes, with which solution?

Answer: The soil fertilizer was weighed and mixed with soil before filling in columns.
Changed in the manuscript: We added a sentence to explain this. The soil was
weighed, mixed perfectly with above fertilizer, respectively, and filled in its own
column.

**Comment 12**: - In figure 2: the authors draw 3 rock tablets, while the authors put only
2 rock tablets at the bottom of the column. Should change it.
Changed in the manuscript: We have changed this.

**Comment 13**: - Did the authors perform the same experiment without rock tablets if
they leach their column in order to observe the leaching solution of the column?
Answer: We didn't design that in this study. We didn't collect the soil solution. The
leaching depended on the rainfall.

**Comment 14**:- Did the authors put the 2 different rock tablets (calcite and dolomite)
in the same column?
Answer: Yes, we did.

**Comment 15**: The authors should explain the reason of the fertilizer weight use in the
experiment.
Answer: Because the added amount of fertilizer can magnify and quicken the
fertilization effect in the short-term according to another experiment from us, and
can't affect the phenomenon we want to observe in this study.   Another paper of ours
(in preparation) about a series of different amount of fertilizer addition will discuss
this issue. The added amounts of these 11 fertilizers were designed only by the
average amount of N, P and K fertilizer in the local practical use.
Changed in the manuscript:   We have added the amount of N, P and K fertilizer in
local practical use in this manuscript like this: N fertilizer: 160 kg N $\cdot$ha$^{-1}$; P fertilizer:
150 kg P$_2$O$_5$ $\cdot$ha$^{-1}$; K fertilizer: 50 kg K $\cdot$ha$^{-1}$)

**Comment 16**: 3. Results - L.164-165: Do not repeat Table 2 and Fig. 3. You may
write: "The results are presented in Table 2 and in Figure 3.
Changed in the manuscript:   We have changed this.

**Comment 17**: 4. Discussion - 4.1.: the first paragraph (L. 182-197) is quite general
and it would be worthy to move it either in the introduction, or at least in the
Materials and Methods section.
Changed in the manuscript: We moved them to the introduction.

**Comment 18**: 4.1. L.213-219: It is exactly the same text as in the introduction (L.
48-54) The authors may express their idea at least a little bit differently.
Answer: Because they are for elaborating different problem, we think we should put
one of them another way.
Changed in the manuscript:   We changed the statements in section 4.2.

**Comment 19**: Information about soils and soil solutions are needed in order to
understand their chemical evolution during the carbonate weathering. - Would it be
possible to present the chemistry of each fertilizer used in this experiment? This can
be added in supplementary information.
Answer:   yes, it is very important. Most of them have been discussed in section 4.2
and 4.3 so far. And we are doing some further research on that.

[revised manuscript text omitted]

Acw - the amount of carbonate weathering; Rcw - the ratio of carbonate weathering; Racw - the rate of carbonate weathering; Acw = (W$i$-W$f$);  Racw = (W$i$-W$f$)/(S*T), where W$i$ is the initial weight of the carbonate rock tablets, and W$f$ is their final weight. S is the surface area of carbonate weathering tablets, and T is the experiment period.

Table 3: The amount of generated $NH_4^+$ at the initial phase of the experiment

| Treatment | Relative molecular mass /g/mol | Amount of added fertilizer /g | Molar concentration /mol | Initial $NH_4^+$ /mol |
|---|---|---|---|---|
| $NH_4NO_3$ | 80 | 43 | 0.54 | 0.54 |
| $NH_4HCO_3$ | 79 | 85 | 1.08 | 1.08 |
| $NaNO_3$ | 85 | 91 | 1.07 | 0.00 |
| $NH_4Cl$ | 53.5 | 57 | 1.07 | 1.07 |
| $(NH_4)_2CO_3$ | 96 | 51 | 0.53 | 1.06 |
| $Ca_3(PO_4)_2$ | 310 | 52 | 0.17 | 0.00 |
| $(NH_4)_3PO_4$ | 149 | 15 | 0.10 | 0.30 |
| Ca-Mg-P | / | 44 | 0.00 | 0.00 |
| Urea | 60 | 32 | 0.53 | 1.06 |
| $K_2CO_3$ | 138 | 10 | 0.07 | 0.00 |

[Figure]

Fig. 1 The change of chemical fertilizer consumption in China during 1980-2013
The data were collected from National Bureau of Statistics of the People's Republic of China
(NBS, 2014) (http://www.stats.gov.cn/tjsj/ndsj/)

[Figure]

         Fig. 2 Sketch map of the soil column

[Figure]

Fig. 3 The ratio of carbonate weathering under different fertilization treatment (a)-limestone; (b)-dolostone. Treatment 1-Control; 2-$NH_4NO_3$; 3-$NH_4HCO_3$; 4-$NaNO_3$; 5-$NH_4Cl$;

6-$(NH_4)_2CO_3$; 7-$Ca_3(PO_4)_2$; 8-$(NH_4)_3PO_4$; 9-Ca-Mg-P; 10-Urea; 11-$K_2CO_3$. $Rcw =(Wi-Wf)/Wi$, where $Wi$ is the initial weight of the carbonate rock tablets, and $Wf$ is their final weight.

[Figure]

Fig. 4 The ratio of limestone weathering and the molar amount of produced $NH_4^+$ under different fertilization treatment

Treatment 1-Control; 2-$NH_4NO_3$; 3-$NH_4HCO_3$; 4-$NaNO_3$; 5-$NH_4Cl$; 6-$(NH_4)_2CO_3$; 7-$Ca_3(PO_4)_2$; 8-$(NH_4)_3PO_4$; 9-Ca-Mg-P; 10-Urea; 11-$K_2CO_3$. R$cw$ =(W$i$-W$f$)/W$i$, where W$i$ is the initial weight of limsestone tablets, and W$f$ is their final weight.

---

## Author Response (AR2)

**Responses and revision descriptions**

Dear Prof. Gerard Govers

Thank you very much for your comments and giving these great suggestions, which make this paper improved a lot.

We have done a major revision according to these suggestions, and the responses and revision descriptions are following below.

I. - The structure of the paper needs improvement. Now, there is still to much mixing of methods, results and discussion. Furrthermore, the results are not all well presented: you report that dolostone and limestone tables were used but only discuss the limestone results because the dolostone results were similar. I would suggest all available data should be presented, at least in tabular form in an appendix and that at least a graph is included showing that the results are indeed similar.

**Changed:** we added some contents of statistical analysis, and analyzed the relationship between dolostone and limestone weathering based on the results of ANOVA analysis.

- With respect to the structure of the paper, the following guidelines may be useful to you:
Introduction: explain the state of the art.
- Weathering may be impacted by mineral fertilization
- But impact of different types of fertilizers not known
- Therefore we conducted experiments

**Changed:** we revised the section like this

3. Materials and methods:
- Description of the experimental set up
- study area, including meteo data and more infor on the soil (grain size Ph, SOC content in a small table)
- columns : size, way of filling, resulting bulk density of the soil.....
- measurement procedure for the tablets and presentation of the way losses are calculated
- the description needs to include a justification of some or your decisions on the methods: why this size of columns ? Why 30 times more fertilizer ? Why did you choose this length for the measuring period ?

**Changed:** we added the description mentioned above, including grain size, bulk density.
The primary reasons why we set up the amount of added fertilizer in this study are: (1) the soil we used is untilled fresh soil which we sampled from B layer. Considering its low nutrition, we set up a higher fertilizer amount. (2)We just want to explore the different response of these different on carbonate weathering, and magnify and quicken the short-term response. (3) It is a simple pre-study, but we think some findings are worthy published especially to $CO_2$ consumption via carbonate weathering at agricultural areas.

Considering its misleading possibility, we decided to delete the relative (30 times) statement.

4. Results
- Presentation of the results: no discussion and no further justification of the study or certain decision taken
- Presentatiion of the weathering rates
- Statistical analysis : for which treatments are rates significantly different: this can be done with an ANOVA analysis. I suggest to include the type of tablet as a class variable here so that we can see whether or not the results for limestone and dolostone are similar

Changed: we added the relative ANOVA analysis.

5. Discussion
- A key element in the discussion are the weathering reactions: I suggest to have the generic weathering reactions in the text (paragraph 4.2) and then have the fertilizer-specific reactions in a table. In this table you can then also indicate the amount of NH4 per mole of fertilizer
- After having done this you can proceed to discuss the differences between the treatments. I think this is already more or less covered in the current version of the MS but it needs to be presented more clearly
- Then you may discuss the fact that CO2 consumption by weaterhing may be wrongly estimated if the contributions of (different) fertilizers is not accounted for
- Finally you have to compare your results with other data: now, there is no quantitative comparison whatsoever with results from other studies. Nevertheless, this is possible: you can calculate a weathering rate from your results and make than reasonable assumptions to make a calculation for larger areas that could be compared to the results of earlier studies whcih you already cite.

Changed: we rewrite the discussion section according to these suggestions above. Thank u again.

6. The English used is not yet up to international standards. The paper really needs a revision by a native speaker

Changed: we changed the mistakes you noted in pdf file and checked the language problems for several times. PLEASE correct it if conveniently.

8. We did the edition and correction in terms of the notes in pdf file.

Thank you SO much for your favor to improve our manuscript.

Best regards,

Song Chao etc.

[revised manuscript text omitted]

Martikainen. 2008. Direct experimental evidence for the contribution of lime to $CO_2$ release from managed peat soil. Soil Biology and Biochemistry 40:2660-2669.

Calmels, D., J. Gaillardet, and L. François. 2014. Sensitivity of carbonate weathering to soil $CO_2$ production by biological activity along a temperate climate transect. Chem Geol 390:74-86.

Chao, S., L. Changli, Z. Yun, and H. Hongbing. 2014. Impact of animal manure addition on agricultural lime weathering in acidic soil: pH dependence and $CO_2$ independence of agricultural lime weathering. Procedia Earth and Planetary Science:405-409.

Chao, S., L. Changli, W. Junkun, Z. Yun, and H. Hongbing. 2011. Impact of the Addition of a Compound Fertilizer on the Dissolution of Carbonate Rock Tablets: a Column Experiment. Appl Geochem 26:170-173.

Chien, S.H., L.I. Prochnow, S. Tu, and C.S. Snyder. 2011. Agronomic and environmental aspects of phosphate fertilizers varying in source and solubility: an update review. Nutr Cycl Agroecosys 89:229-255.

Chou, L., R.M. Garrels, and R. Wollast. 1989. Comparative study of the kinetics and mechanisms of dissolution of carbonate minerals. Chem Geol 78:269-282.

Dreybrodt, W. 1988. Processes in karst systems: physics, chemistry, and geology Springer-Verlag Berlin Heidelberg.

Dreybrodt, W., J. Lauckner, L. Zaihua, U. Svensson, and D. Buhmann. 1996. The kinetics of the reaction $CO_2 + H_2O \rightarrow H^+ + HCO_3^-$ as one of the rate limiting steps for the dissolution of calcite in the system $H_2O$-$CO_2$-$CaCO_3$. Geochim Cosmochim Ac 60:3375-3381.

Egli, M., and P. Fitze. 2001. Quantitative aspects of carbonate leaching of soils with differing ages and climates. Catena 46:35-62.

Errin, A.S.P., A.P. Robst, and J.L.P. Robst. 2006. Impact of nitrogen fertilizers on natural weathering processes : Evident role on $CO_2$ consumption The oxygen isotopic composition of Precambrian cherts, pp. 2332-2332 Goldschmidt Conference, Vol. 58.

Etchanchu, D., and J. Probst. 1988. Evolution of the chemical composition of the Garonne River water during the period 1971-1984. Hydrological sciences journal 33:243-256.

(ed.) 1981. Proceedings of the 8th International Congress of Speleology Bowling Green, Kentucky.

Gams, I. 1985. International comparative measurements of surface solution by means of standard limestone tablets. Razpr IV Razeda Sazu 26:361-386.

Gandois, L., A.S. Perrin, and A. Probst. 2011. Impact of nitrogenous fertiliser-induced proton release on cultivated soils with contrasting carbonate contents: A column experiment. Geochim Cosmochim Ac 75:1185-1198.

GBS. 2014. 2014 Guizhou Statistical Yearbook Statistical Press of China, Beijing.

Hagedorn, B., and I. Cartwright. 2009. Climatic and lithologic controls on the temporal and spatial variability of $CO_2$ consumption via chemical weathering: An example from the Australian Victorian Alps. Chem Geol 260:234-253.

Hamilton, S.K., A.L. Kurzman, C. Arango, L. Jin, and G.P. Robertson. 2007. Evidence for carbon sequestration by agricultural liming. Global Biogeochem Cy 21:1-12.

Han, Z., and Z. Jin. 1996. Hydrogeology of Guizhou Province, China Seismic Publication, Beijing.

Hunter, A.H. 1980. Laboratory and greenhouse techniques for nutrient survey to determine the soil amendments required for optimum plant growth. Mimeograph. Agro Service International (ASI), Florida, USA.

Jiang, Y. 2013. The contribution of human activities to dissolved inorganic carbon fluxes in a karst underground river system: Evidence from major elements and $\delta^{13}C_{DIC}$ in Nandong, Southwest China. J Contam Hydrol 152:1-11.

Jiang, Y., Y. Wu, C. Groves, D. Yuan, and P. Kambesis. 2009. Natural and anthropogenic factors affecting the groundwater quality in the Nandong karst underground river system in Yunan, China. J Contam Hydrol 109:49-61.

Jiang, Z., and D. Yuan. 1999. $CO_2$ source-sink in karst processes in karst areas of China. Episodes 22:33-35.

Jiang, Z., Y. Lian, and X. Qin. 2013. Carbon cycle in the epikarst systems and its ecological effects in South China. Environ Earth Sci 68:151-158.

Kiefer, R.H. 1994. Temporal cycles of karst denudation in northwest Georgia, USA. Earth Surf Proc Land 19:213-232.

Kitano, Y., M. Okumura, and M. Idogaki. 1978. Uptake of phosphate ions by calcium carbonate. Geochem J:29-37.

Kump, L.R., S.L. Brantley, and M.A. Arthur. 2000. Chemical weathering, atmospheric $CO_2$, and climate. Annu Rev Earth Pl Sc 28:611-667.

Lang, Y., C. Liu, Z. Zhao, S. Li, and G. Han. 2006. Geochemistry of surface and ground water in Guiyang, China: Water/rock interaction and pollution in a karst hydrological system. Appl Geochem 21:887-903.

Lerman, A., and L. Wu. 2006. $CO_2$ and sulfuric acid controls of weathering and river water composition. J Geochem Explor 88:427-430.

Lerman, A., L. Wu, and F.T. Mackenzie. 2007. $CO_2$ and $H_2SO_4$ consumption in weathering and material transport to the ocean, and their role in the global carbon balance. Mar Chem 106:326-350.

Li, S., D. Calmels, G. Han, J. Gaillardet, and C. Liu. 2008. Sulfuric acid as an agent of carbonate weathering constrained by $\delta^{13}C_{DIC}$: Examples from Southwest China. Earth Planet Sc Lett 270:189-199.

Li, S., Z. Xu, H. Wang, J. Wang, and Q. Zhang. 2009. Geochemistry of the upper Han River basin, China: 3: Anthropogenic inputs and chemical weathering to the dissolved load. Chem Geol 264:89-95.

Lian, B., Y. Chen, L. Zhu, and R. Yang. 2008. Effect of Microbial Weathering on Carbonate Rocks. Earth Science Frontiers 15:90-99.

Liu, C., S. Li, Y. Lang, and H. Xiao. 2006. Using $\delta^{15}N$- and $\delta^{18}O$-Values To Identify Nitrate Sources in Karst Ground Water, Guiyang, Southwest China. Environ Sci Technol 40:6928-6933.

Liu, Z. 2011. " Method of maximum potential dissolution" to calculate the intensity of karst process and the relevant carbon sink: With discussions on methods of solute load and carbonate-rock-tablet test. Carsologica Sinica 30:79-82.

Liu, Z., and W. Dreybrod. 1997. Dissolution kinetics of calcium carbonate minerals in $H_2O$-$CO_2$ solutions in turbulent flow: The role of the diffusion boundary layer and the slow reaction $H_2O + CO_2 \leftrightarrow H^+ + HCO_3^-$. Geochim Cosmochim Ac 61:2879-2889.

Liu, Z., W. Dreybrodt, and H. Liu. 2011. Atmospheric $CO_2$ sink: silicate weathering or carbonate weathering? Appl Geochem 26:292-294.

Liu, Z., H. Sun, L. Baoying, L. Xiangling, Y. Wenbing, and Z. Cheng. 2010. Wet-dry seasonal variations of hydrochemistry and carbonate precipitation rates in a travertine-depositing canal at Baishuitai, Yunnan, SW China: Implications for the formation of biannual laminae in travertine and for climatic reconstruction. Chem Geol 273:258-266.

NBS. 2014. 2014 Statistical Yearbook. Statistical Press of China, Beijing.

Oh, N., and P.A. Raymond. 2006. Contribution of agricultural liming to riverine bicarbonate export and $CO_2$ sequestration in the Ohio River basin. Global Biogeochem Cy 20:1-17.

Perrin, A., A. Probst, and J. Probst. 2008. Impact of nitrogenous fertilizers on carbonate dissolution in small agricultural catchments: Implications for weathering $CO_2$ uptake at regional and global scales. Geochim Cosmochim Ac 72:3105-3123.

[revised manuscript text omitted]

---

## Author Response (AR3)

**Reviewer 1**

1. In some paragraphs, the English writing may be slightly improved.
**Changed in the manuscript:** we have re-edited the language question by the

Language Services of Elsevier.

2. In the field, how are placed the different experimental columns? Did you have a random position, mixing the different modalities?
**Answer:** we placed orderly the different treatments columns including replicated treatments.
**Changed in the manuscript:** we added "were labelled and placed orderly" to interpret this.

3. The authors mixed "perfectly" the chemical fertilizers with the sieved soils. It was not directly spread on the field/columns. This may be an artefact compared to the natural field, where fertilizers are spread on the soils, relatively far from the rock.
**Answer:** We agree that this design have difference from the natural field. But considering the uniformity of all treatments, the results we got are still convinced. We will consider the comparison research in terms of different way and amounts of fertilizer addition.

4. In the last part of the discussion, this point may be highlighted, as the authors compared their data with the literature… This way, their weathering rate may be (slightly) overestimated. This other point is that in the literature most studies approach the weathering estimate from riverine data as the authors discussed.
**Answer:** we are pleased that you agree with that.

‑ L41 – 42 : "(…) processes including the reaction between carbonates and  protons derived (…)".
**Changed in the manuscript:** We removed " the".

‑ L 45 and 47: you may be more specific on the origin of sulfuric acid, and the role of acidic soil in the carbonate – proton relationship. "Acidic soil" is still too broad.

**Changed in the manuscript:** changing into **"**sulfuric acid forming in the oxidation of reduced sulfuric minerals (mainly pyrite, $FeS_2$),……. acidic soil (such as red soil, yellow soil) "

‑ L55, you may add the increase proportion of mineral fertilizers (increase by 365%), in order to compare with the 3.3% worldwide increase… I was wondering what was the cause of such sharp increase in chemical fertilizers consumption. Is it just an effect of the increase of the size of agricultural land or is it a consequence of a change of fertilizer habit (more NO3 or NH4 fertlizers)?

**Answer:** The primary reasons why we set up the amount of added fertilizer in this study are: (1) the soil we used is untilled fresh soil which we sampled from B layer. Considering its low nutrition, we set up a higher fertilizer amount. (2)We just want to explore the different response of these different on carbonate weathering, and magnify and quicken the short-term response. (3) It is a simple pre-study, but we think some findings are worthy published especially to $CO_2$ consumption via carbonate weathering at agricultural areas.

8－L118－119: You worked on Guizhou area where the consumption of chemical fertilizers increased by about 26% (far less than the increase of fertilizer consumption at the scale of the whole country).
**Answer:** It is true that according to the 2014 Guizhou statistical yearbook.

9－L121: Could you be more specific for the soil classification, more precise than B horizon?
**Changed in the manuscript:** we added "yellow-brown clay".

10－L123: did you crush (ground) the sample before to pass it through 2 mm sieve? Or did you pass the air－dried soil through 2 mm sieve, and after you crushed it… this may seem to be a detail, but it is really important.
**Answer:** all of soil samplers we sampled were ground first, and then sieved.

11－L141: Did you have any silt or clay loss with a 0.5 mm net at the bottom of your column? Did you add some quartz wool in addition to the sand "filter" above the PE net?
**Answer:** we didn't consider the clay loss with the smaller particle. We will consider the quartz wool material as the member of filters in future studies. Thank you so much for your suggestions.

12－L142－143: "Two different carbonate rock tablets were buried in the bottom of each soil column"; by this, do you mean that you put only one tablet of two different carbonate rocks? So the carbonate rocks are different, or are they two aliquots of the same rock type?
**Answer:** yes, we placed only one tablet of each carbonate rock in one column, but we designed 3 columns for each treatment as replicates. The carbonate rocks are different absolutely got from different area.

13－L145－150: How did you determine the weight of each chemical fertilizer applied on each soil experiment?
**Answer:** The primary reasons why we set up the amount of added fertilizer in this study are: (1) the soil we used is untilled fresh soil which we sampled from B layer. Considering its low nutrition, we set up a higher fertilizer amount. (2)We just want to explore the different response of these different on carbonate weathering, and magnify and quicken the short-term response. (3) It is a simple pre-study, but we think some findings are worthy published especially to $CO_2$ consumption via carbonate weathering at agricultural areas.

In order to short the time of this experiment, and also considering the low nutrients of the fresh soil, we have added the amount of N, P and K fertilizer in local practical use in this manuscript like this: (N fertilizer: 160 kg N·ha-1; P fertilizer: 150 kg P2O5·ha-1; K fertilizer: 50 kg K·ha-1). According to the chemical formula and the molecular weight of each fertilizer, we finally calculated the amount of added fertilizer for each treatment.

– L151 – 152: you did mix each fertilizer with air – dried soils. So the fertilizer was not spread on the column? As soil and fertilizers were perfectly mixed, did you also crush the chemical fertilizers or did you leave them in their original shapes?
**Answer:** All the fertilizers are small granular with homogeneous size. They were perfectly mixed with soil. Undeniably speaking, the fertilizer may be spread partly on the wall of column in some cases, but we believe that its effect on our experiment results can be ignored. We will think it about in our further researches

– L158 – 159: you used 2 carbonate types: a limestone and a dolostone. Did you put one tablet of each carbonate in the same column, or did you put 2 tablets of the same carbonate (in order to have duplicates) in the same column?
– L176: This is at this point that we know that you have triplicates. How did you obtain these triplicates: did you use different columns for replicates, or are they in the same columns… It is important in order to understand on which data you performed a statistical analysis. How many columns did you in total?
**Answer for 15 and 16:** we placed only one tablet of each carbonate rock in one column, but we designed 3 columns for each treatment as replicates.

– You should add "respectively" at the end of your sentence when you're listing some results…
**Changed in the manuscript:** we added it.

– L189: the "rest treatments"? do you mean the other chemical fertilizers other than NH4NO3, NH4Cl, (NH4)2CO3, NH4HCO3?
**Answer:** yes, it is pointing the other chemical fertilizers than urea, NH4NO3, NH4Cl, (NH4)2CO3, NH4HCO3.
**Changed in the manuscript:** The English language editor in Elsevier suggested us changing it into the remaining treatment, we did it.

– L206: you explain that there is no difference in the weathering behavior of limestone and dolostone, however the control experiment show different behavior. Did you test this similarity?
**Answer:** yes, it is true that there is a difference between limestone and dolostone in control treatment. But we plotted the Rw of limestone vs. dolostone tablets in a linear correlation diagram, the results show that the Rw of limestone and dolostone exhibit a high positive correlation ($R^2=0.9773$; see Fig. 4), indicating that the weathering of limestone and dolostone are similar under different treatments.
**Changed in the manuscript:** see the details in the manuscript.

– L213 – 215: you may present these 3 equations in the following order: (5) > (4) > (3), following an increasing pKa (or pH).
**Changed in the manuscript:** we changed it.

– L218: H2CO3 is not only formed in the soil, it is first formed in the atmosphere with the dissolution of atmospheric CO2(g) into rain droplets and rain, as CO2aq and then H2CO3. The concentration of H2CO3 is exacerbated into soil because of the presence of organic CO2 from respiration. And yes, one of the main control of carbonate dissolution is the amount of rainfall.

That's why is it important to know how the different columns were placed in the field, randomly or the same modalities were placed together (L223–224). If randomly, you can say that rainfall may not be consider a controlling effect for your experiment.

**Answer:** Thank you very much. It is true that the rainfall is a main controlling factor in the natural weathering processes. The rate of chemical weathering is higher in high-precipitation area. However, in this study, we placed orderly the different treatments columns including replicated treatments in one specific place of field. To each treatment, the precipitation is same. So we conclude that the difference in weathering rate in each treatment is not caused by the precipitation. That's what we are meaning.

**Changed in the manuscript:** We changed it to make it clearer, see the manuscript.

– L231–232: you may be more precise for "$CO_2$ degassing"… Issued from carbonate dissolution? For respiration?

**Answer:** In theory, the source of $CO_2$ dissolved in water is from respiration (common cases), deep crust (tufa formation; areas where have thermal mineral spring), artificial source (CO2 capture and storage), etc. It is another big problem. No matter where they come from, these $CO_2$ usually keep in their balance status in specific stratum. Some of them keep their balance in $H_2O$-$CO_2$-$HCO_3$-carbonate system in water. Dramatic changes in the parameters of the $CO_2$ system such as T, pH and/or $pCO_2$ can cause $CO_2$ degassing. Here, we just listed some cases which can result in $CaCO_3$ precipitation.

**Changed in the manuscript:** we added some information like this "the degassing of dissolved $CO_2$ due to dramatic changes in the parameters of the $CO_2$ system (such as T, pH, $pCO_2$, etc)"

– L244: the year of the publication from Singh et al is missing.
**Changed in the manuscript:** it was added.

– L305: you did calculate the initial fertilizer –derived NH4 per unit. But it would be interesting to have the initial rate of fertilizer spread in the field and to compare it with what it is really applied in the Chinese agricultural watersheds.

**Answer:** The reason why Table 4 was given is mainly to better understand the N balance of these fertilizers in their own reactions listed in table 3, to interpret how many $NO_3$ derived from nitrification to further evaluate the effect of fertilizer on carbonate weathering.

Yes, we believe that the comparison between the addition amount of fertilizer in this study and that in practical agricultural activity of local area is very interesting question. But it is another interesting question. It has a little difficult to do it here and make it sense since it needs another experiment design. We will consider conducting some experiment to fill the gaps between them in the future.

– L396–397: – Table 2: what is the significance of a, b, c behind the numbers (ANOVA)?

**Answer:** we deleted them and plotted the Rw of limestone vs. dolostone tablets in a linear correlation diagram ($R^2$=0.9773; see Fig. 4) instead to further explain it. See the manuscript.

**Reviewer 2**

1. In the methods section, very few data seem to have been collected from the soils during the experiment. For example was pH monitored? This is an important parameter and would have helped the authors determine which mechanism is responsible for their results.

**Answer:** Before experiment, we tested the parameter of soil we used including soil pH organic matter (OM), $NH_4$-N, $NO_3$-N, available P, available K, available Ca, available Mg, available Fe, and available S. The results were listed in Table 1.
As you mentioned, we regret pretty much that we didn't monitor their change by testing them after experiment.

2. The authors carry out experiments on soil columns in a field setting. What exactly does this mean? Some photos of the experimental setup might help explain.
 **Answer:** The shape and inner structure of columns in this study were described in Fig. 2. We orderly placed them (11treatments*3 triplicates= 33 columns) in field (on line and row). No suitable pictures here.

3. The authors should discuss the role played by bacteria in the system. Presumably the fertilizers could have stimulated bacteria which may have increased respiration and CO2 concentrations in the soil. This might account for some of the enhanced weathering. Can this mechanism be assessed?

**Answer:** Thank you for your comment on this, the probability truly exits more or less here, but we think the effect on nitrification is more distinct according to the results in the urea, $NH_4NO_3$, $NH_4HCO_3$, $NH_4Cl$, and $(NH_4)_2CO_3$ treatments. It is very difficult to distinguish and quantify their contribution in this study. We may try to do it by some isotope methods.

**Changed in the manuscript:** we added relative statements in Section 4.1.
In theory, the fertilizers could stimulate bacteria, which may increase respiration and $CO_2$ concentrations in the soil, as a result, probable enhance carbonate weathering as Eq. (5).

4. There are no high resolution SEM images of the tablets before or after the experiment. This may help the authors understand the loss mechanisms a bit better. The authors focus solely on chemical dissolution, although recently it has been suggested that mechanical grain detachment in carbonate rocks is likely to be an important weathering pathway. Imaging might help identify such a process.

**Answer:** Your suggestion about SEM is very inspiring for us. We will use this method in our future study. We appreciate.

5. Why was the reason for selecting the specific amounts of fertilizer? There is no explanation at all. Does this correspond to the amounts typically applied to crops? Also if this is to simulate the agricultural application does it make sense to mix the fertilizer thoroughly into the soil?

**Answer:** The primary reasons why we set up the amount of added fertilizer in this study are: (1) the soil we used is untilled fresh soil which we sampled from B layer. Considering its low nutrition, we set up a higher fertilizer amount. (2)We just want to explore the different response of these different on carbonate weathering, and magnify and quicken the short-term response. (3) It is a simple pre-study, but we think some findings are worthy published especially to $CO_2$ consumption via carbonate weathering at agricultural areas.

The amount is just for this comparison experiment. Undeniably, it is not perfect for linking this experiment with practical agricultural activities in local area. We are fixing on that. Some data and papers are in preparation.

6. What kind of limestone and dolostone were used? What was the grain size and composition for example? Again, some SEM images here would help.
**Answer:** As described in Methods section, the statement on carbonate rock tablets used in this study is that "Two different kinds of carbonate rock tablets (2 cm $\times$ 1 cm $\times$ 0.5 cm in size) were placed in the bottom of each soil column to examine the rate of carbonate weathering in the soil. The two different kinds of carbonate rock collected from the karst area of Huaxi District were: (1) limestone with 60-65 % micrite, 30-35 % microcrystalline calcite, and 2-3 % pyrite; and (2) dolostone with 98-99 % fine crystalline dolomite, 1 % pyrite, and trace quantities organic matter."

SEM image are a good advice for our further study. Thank you so much.

7. Although the authors use both limestone and dolostone they don't really discuss the differences in detail. For example, on the whole dolostone seems to weather more slowly than limestone, but there are some exceptions (ammonium carbonate for instance). Are the differences significant? If so, what do they mean?
**Answer:** Thank you for your comment on it. We note it and added some statement on it. But there are no more points to discuss further, we think.

**Changed in the manuscript:** We added the following statements in Section 3.2.
Fig. 3 shows that, on the whole, the ratios of dolostone weathering are smaller than those of limestone weathering except $(NH_4)_2CO_3$ treatment, exhibiting that dolostone weather more slowly than limestone under fertilization effects.
In Fig. 4, we plotted the R$w$ of limestone vs. dolostone tablets in a linear correlation diagram, in order to compare the weathering responses of limestone with dolostone. The results show that the R$w$ of limestone and dolostone exhibit a high positive correlation ($R^2$=0.9773; see Fig. 4), indicating that the weathering of limestone and dolostone are similar under different treatments. Thus, we will explain the results in terms of carbonates, rather than by way of the individual dolostone and limestone.

8. In the conclusions, the authors state that the fact that the ammonium phosphate and sodium nitrate treatments did not impact the weathering rate raises a new question. But they don't explicitly state what this question is. A similar phrase is used in the Abstract.
**Changed in the manuscript:** we noticed that and made it clearer, we used a colon instead of a full stop in Abstract section:
The results of $NaNO_3$ treatment raise a new question: the negligible impact of nitrate on carbonate weathering may result in overestimation of the impact of N-fertilizer on $CO_2$ consumption by carbonate weathering at the regional/global scale, if the effects of $NO_3$ and $NH_4$ are not distinguished.
In conclusion part: we added a pointing expression like: The question is: ……..

9. Figure 3 is a bit confusing because of all the lower case letters (a,b, bc etc) dotted all over. What do they mean? It's not explained in the caption. **Changed in the manuscript:** we deleted them and plotted the Rw of limestone vs. dolostone tablets in a linear correlation diagram (R2=0.9773; see Fig. 4) instead to further explain it. See the manuscript.

10. Finally, careful English language editing would significantly improve the paper. Apart from numerous grammatical errors, it may also help organize the discussion. For example the section 4.2 is a bit of a mess and does not flow at all. **Changed in the manuscript:** we have re-edited the language question by the

Language Services of Elsevier.

[revised manuscript text omitted]
 fine crystalline dolomite, 3-5% microcrystalline calcite, 1 % pyrite, and little trace quantities organic matter. All of the tablets were baked heated at 80 °C for 4 hours, then weighed in a 1/10000 electronic balance in the laboratory, tied to a labeled by tying a label with fishing line, and then buried at the bottom of each soil column. After a whole year, Tthey tablets were taken outremoved carefully, rinsed, baked and weighed after a whole year.

The amount of weathering carbonate weathering (A$ew$), the ratio of carbonate weathering (R$ew$) and the rate of carbonate weathering (R$aew$) for limestone and dolomite were calculated according to the weight difference of the tablets using the following formulas:

$$A_{ew} = (W_i - W_f) \tag{1}$$

$$R_{ew} = (W_i - W_f)/ W_i \tag{2}$$

$$R_{acw} = (W_i\text{-}W_f)/(S*T) \qquad (3)$$

where W$i$ is the initial weight of the carbonate rock tablet, W$f$ is the final weight,

S is the surface area of carbonate  rock tablet, and T is the length of the experiment period.

**2.5 Statistical analysis**

Statistical analysis was performed using IBM SPSS 20.0 (Statistical Graphics Corp, Princeton, USA). All results of carbonate weathering were reported as the means ± standard deviations (SD) for the three replications.

**3. Results**

3.1 Weathering rate of carbonate under different fertilized treatments

The  and  limestone and dolostoneare listed  The results showed that the A$_{cw}$,  limestone  under urea, NH$_4$NO$_3$, NH$_4$Cl, (NH$_4$)$_2$CO$_3$and NH$_4$HCO$_3$ treatments were 8.48 ± 0.96, 6.42 ± 0.28, 5.54 ± 0.64, 4.44 ± 0.81 and 4.48 ± 0.95 ‰ , respectively, significantly greater than that under the control treatment 0.48 ± 0.14 ‰ (see Fig. 3). In addition, the observed R$w$ of

 dolomitstone were $6.59 \pm 0.67$, $5.30 \pm 0.87$, $4.77 \pm 0.78$, $4.94 \pm 1.91$ and $3.22 \pm 0.87$ ‰ respectively, under these same five fertilization treatments, in contrast to  $-0.31 \pm 0.09$ ‰ in the control treatment. This clearly demonstrates that the addition of these five fertilizers  increase the rate of  carbonate weathering.

The remaining  treatments made no significant differences in the $R_w$ and $R_{aw}$ of limestone and dolomitstone in comparison  to the control treatment (Fig. 3).  In the $(NH_4)_3PO_4$ treatment, the $R_w$ were only $1.08 \pm 0.34$ ‰, $0.75 \pm 0.21$‰ for limestone and dolomite, respectively, while the $R_{aw}$ were $4.00 \pm 1.15$ $g \cdot m^{-2} \cdot a^{-1}$  and $1.00 \pm 1.01$  $g \cdot m^{-2} \cdot a^{-1}$  for limestone and dolomite, respectively. These values are less than those under the other four $NH_4$-fertilizers, as mentioned above. The  $R_w$ and $R_{aw}$ in the $NaNO_3$ treatment failed to show  notable difference with the control treatment,  exhibiting little effect of the $NaNO_3$ fertilizer addition on carbonate weathering (Fig. 3). –

Except for the $R_{cw}$ of limestone approaching zero in the $Ca_3(PO_4)_2$ treatment , all the values of  $R_{cw}$ and $R_{aw}$  in Ca-Mg-P , $K_2CO_3$ and $Ca_3(PO_4)_2$ treatments showed  negative value, This  indicates that the addition of Ca-Mg-P, $K_2CO_3$ and $Ca_3(PO_4)_2$ fertilizers led to  precipitation at the surface of the carbonate mineral, which can be explained by common ion effect.

3.2 Comparison of limestone of dolomite

Fig. 3 shows that, on the whole, the ratios of dolostone weathering are smaller than those of limestone weathering except $(NH_4)_2CO_3$ treatment, exhibiting that dolostone weather more slowly than limestone under fertilization effects.

In Fig. 4, we plotted the R*w* of  limestone vs. dolostone tablets in a linear correlation diagram, in order to compare the weathering responses of limestone with dolostone. The results show that the R*w* of limestone and dolostone exhibit a high positive correlation (R$^2$=0.9773; see Fig. 4), indicating that the weathering of  limestone and dolostone are similar under different treatments. Thus, we will explain the results  in terms of carbonates, rather than by way of the individual dolostone and limestone.

**4. Discussion**

~~**The carbonate rock tablet test is used to determine the weathering rate of carbonate rock/mineral from laboratory to field (Gams, 1981; Chao et al., 2011; Trudgill, 1975; Chao et al., 2014; Dreybrodt et al., 1996; Gams, 1985; Jiang and Yuan, 1999; Liu and Dreybrod, 1997; Plan, 2005). In laboratory, the carbonate rock tablet is employed to study the kinetics of calcite dissolution/precipitation (Dreybrodt et al., 1996; Liu and Dreybrod, 1997) and~~

determine the rate of carbonate mineral weathering in soil column (Chao et al., 2011). However, in field, it is also used to observe the rate of carbonate weathering and estimated $CO_2$ consumption by carbonate weathering (Chao et al., 2014; Jiang and Yuan, 1999; Jiang et al., 2013; Plan, 2005). Although Liu (2011) argue that the carbonate rock tablet test may lead to the deviation of estimated $CO_2$ consumption by carbonate weathering at the regional/global scale in the case of insufficient representative data (Liu, 2011), our results show that it is a preferred option for the condition controlled contrast or stimulated experiment (Chao et al., 2011; Chao et al., 2014), where the result from the carbonate-rock-tablet test is consistent to the major element geochemical data of leachates from soil column(Chao et al., 2011).

In this study, every procedure to establish soil column with carbonate rock tablets in the bottom of each was strictly same, including the size of column, the preparation and column filling of soil sample, the setting and test of carbonate rock tablets, etc. Moreover, three replicates of each treatment were designed. We consider the experiment design can meet the objective of this study and the results of carbonate-rock-tablet test are therefore valid and credible.

**4.1 The kKinetics of carbonate dissolution/precipitation: controlling factors**

Experimental studies of carbonate dissolution kinetics have shown metal carbonate weathering usually depends upon three parallel reactions occurring at the carbonate interface (Chou et al., 1989; Plummer et al., 1978; Pokrovsky et al., 2009)(Chou et al., 1989; Plummer et al., 1978; Pokrovsky et al., 2009):

$$MeCO_3 \leftrightarrow Me^{2+} + CO_3^{2-} \tag{4}$$

$$MeCO_3 + H_2CO_3 \leftrightarrow Me^{2+} + 2HCO_3^- \tag{5}$$

$$MeCO_3 + H^+ \leftrightarrow Me^{2+} + HCO_3^- \tag{46}$$

$$MeCO_3 + H_2CO_3 \leftrightarrow Me^{2+} + 2HCO_3^- \tag{5}$$

$$MeCO_3 \leftrightarrow Me^{2+} + CO_3^{2-} \tag{6}$$

where Me = Ca, Mg. As Eq. (5) describes, atmospheric/soil $CO_2$ is usually regardconsidered to be as the natural weathering agent of carbonate. In watersheds with calcite- and dolomite-containing bedrock, $H_2CO_3$ formed in the soil zone usually reacts with carbonate minerals, resulting in dissolved Ca, Mg, and $HCO_3^-$ as described in Eq. (5) (Andrews and Schlesinger, 2001; Shin et al., 2014)(Andrews and Schlesinger, 2001; Shin et al., 2014). Although it has been proven that the reaction of carbonate dissolution is mainly controlled by the amount of rainfall (Amiotte Suchet et al., 2003; Egli and Fitze, 2001; Kiefer, 1994)(Amiotte Suchet et al., 2003; Egli and Fitze, 2001; Kiefer, 1994), in this study, we consider that the effect of rainfall is equal in each soil column, and hence is disregarded unconsidered as a controlling factor in weathering rate differences among these treatmentsin this study. In theory, the fertilizers could stimulate bacteria, which may increase respiration and $CO_2$ concentrations in the soil,      as a result, probable enhance carbonate weathering as Eq. (5). However, The Eq. (46) suggests that the proton from other origins, such as the nitrification processes of $NH_4^+$, as mentioned in the iIntroduction section, can play the role of weathering agent in agricultural areas. In this study, the urea, $NH_4NO_3$, $NH_4HCO_3$, $NH_4Cl$, and $(NH_4)_2CO_3$ amendments increased (10 to 17-fold) the natural weathering rate offrom 2.00 $g \cdot m^{-2} \cdot a^{-1}$ fromfor limestone tablets in the control treatment (tTable 2). Thus, these increases are strongly relativeted 
[revised manuscript text omitted]
 Ccalcium Pphosphate Nnucleation and Ggrowth on Ccalcite: Implications for Ppredicting the Ffate of Ddissolved Pphosphate Sspecies in Aalkaline Ssoils. Environmental Science & Technology, 46(2): 834-842.

West, T.O. and McBride, A.C., 2005. The contribution of agricultural lime to carbon dioxide emissions in the United States: dissolution, transport, and net emissions. Agriculture, Ecosystems and Environment, 108(2): 145-154.

Yue, F.J., Li, S.L., Liu, C.Q., Lang, Y.C. and Ding, H., 2015. Sources and transport of nitrate constrained by the isotopic technique in a karst catchment: an example from Southwest China. Hydrological Processes, 29(8): 1883-1893.

Zeng, C., Zhao, M., Yang, R. and Liu, Z., 2014. Comparison of karst processes-related carbon sink intensity calculated by carbonate rock tablet test and solute load method: a case study in the Chenqi karst spring system. Hydrogeology & Engineering Geology, 41(1): 106-111.

Zhang, C., 2011. Carbonate rock dissolution rates in different landuses and their carbon sink effect.

Chinese Science Bulletin, 56(35): 3759-3765.

[revised manuscript text omitted]